# LongProc: Benchmarking Long-Context Language Models on Long Procedural Generation

**Xi Ye**♠  **Fangcong Yin**◇*  **Yinghui He**♠*  **Joie Zhang**♠*  **Howard Yen**♠*
**Tianyu Gao**♠  **Greg Durrett**◇  **Danqi Chen**♠
♠ Princeton Language and Intelligence   ◇ The University of Texas at Austin
xi.ye@princeton.edu

## Abstract

Existing benchmarks for evaluating long-context language models (LCLMs) primarily focus on long-context recall, requiring models to produce short responses based on a few critical snippets while processing thousands of irrelevant tokens. We introduce LONGPROC (**Long Proc**edural Generation), a new benchmark that requires both the integration of highly dispersed information and long-form generation. LONGPROC consists of six diverse procedural generation tasks, such as extracting structured information from HTML pages into a TSV format and executing complex search procedures to create travel plans. These tasks challenge LCLMs by testing their ability to follow detailed procedural instructions, synthesize and reason over dispersed information, and generate structured, long-form outputs (up to 8K tokens). Furthermore, as these tasks adhere to deterministic procedures and yield structured outputs, they enable reliable rule-based evaluation. We evaluated 23 LCLMs, including instruction-tuned models and recent reasoning models, on LONGPROC at three difficulty levels, with the maximum number of output tokens set at 500, 2K, and 8K. Notably, while all tested models claim a context window size above 32K tokens, open-weight models typically falter on 2K-token tasks, and closed-source models like GPT-4o show significant degradation on 8K-token tasks. Reasoning models achieve stronger overall performance in long-form generation, benefiting from long CoT training. Further analysis reveals that LCLMs struggle to maintain long-range coherence in long-form generations. These findings highlight critical limitations in current LCLMs and suggest substantial room for improvement.[1]

## 1 Introduction

Recent research has expanded the context window of pretrained language models from hundreds of tokens (Devlin et al., 2019; Raffel et al., 2020) to millions (Anthropic, 2024; Gemini Team, 2023), driven by advances in pre-training (Llama-3 Team, 2024), architectures (Gu & Dao, 2024; Sun et al., 2024), and data engineering methods (Fu et al., 2024; Gao et al., 2024). This rapid growth in context window capacity presents new challenges for evaluating these long-context language models (LCLMs). A majority of existing benchmarks focus on tasks that require processing long inputs while producing relatively short outputs, such as answering a simple question or generating a short summary (Kamradt, 2023; Hsieh et al., 2024; Yen et al., 2025). Moreover, these benchmarks typically only test *low-dispersion* scenarios (Goldman et al., 2024), requiring LCLMs to locate and use a few relevant snippets within the long contexts. Such evaluations provide limited insight into LCLMs' performance on more practical long-context tasks that require integration of dispersed information and long-form generation, such as a web agent that synthesizes information across multiple pages while generating lengthy trajectories.

---

*Authors contributing to dataset generation.
[1]Data and code available at: https://princeton-pli.github.io/LongProc.

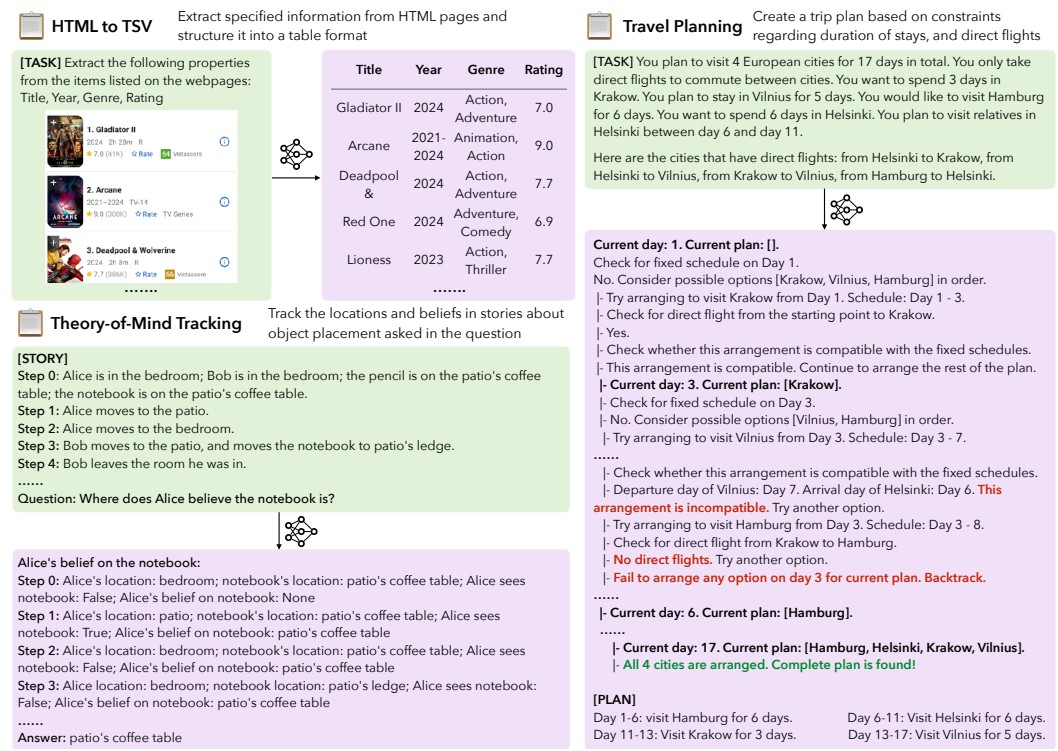

Figure 1: Three (out of six) representative tasks in LONGPROC: HTML to TSV, Theory-of-Mind Tracking, Travel Planning. Each task requires LCLMs to follow a given procedure and generate outputs in a **specified format** detailed by instructions. This leads to natural long-form outputs that can be evaluated reliably with rule-based metrics. Refer to appendix F for examples of all six tasks.

Given the limitations of existing benchmarks, we propose a new paradigm for evaluating LCLMs through **long procedural generation**, which requires models to follow specified procedures and generate structured outputs. As shown in Figure 1, example tasks include extracting target information from lengthy unstructured HTML documents into structured TSV files, or creating trip plans by executing prolonged search procedures. Because these procedures involve multiple generation steps, each requiring LCLMs to use information from the context and previous steps, such tests naturally require integration of dispersed information and long-form generation.

Based on this design principle, we introduce **LONGPROC** (**Long Proc**edural generation), a new benchmark consisting of six distinct tasks (see Table 2 for details). This set of tasks poses diverse challenges in accessing information, multi-step reasoning, and complex search procedures, with direct relevance to practical applications such as web agents (Li et al., 2024c; Zhou et al., 2023) and real-world planning tasks (Zheng et al., 2024; Xie et al., 2024). The required ability for long-context reasoning serves as a foundation to recent advances demonstrated by OpenAI o1/o3 (OpenAI, 2024) and DeepSeek-R1 (Guo et al., 2025) models. In addition, since all tasks follow deterministic procedures and produce structured outputs, they enable reliable rule-based evaluation of extended generations.

LONGPROC features **three difficulty levels** by categorizing the task instances based on required output lengths: 500, 2K, and 8K tokens. These levels effectively differentiate the capabilities of models across different families and scales. We evaluate 23 LCLMs, covering both instruction-tuned models and recent reasoning models. Our results reveal key limitations in current LCLMs' capabilities for long procedural generation: all tested models, including frontier proprietary models, largely fail at 8K-token tasks despite their advertised 128K context windows. Comparisons between reasoning models and instruct models sug-

gest benefits of long CoT training for long procedural generation tasks. Further analysis demonstrates that that models make more mistakes in later generation segments, highlighting models' challenges in maintaining long-range coherence. In summary, LONGPROC serves as a testbed for evaluating LCLMs, exposing limitations that emphasize substantial room for future research in long-context modeling.

## 2 Existing Benchmarks for Evaluating LCLMs

Table 1 reviews recent efforts in developing benchmarks for LCLMs and discuss their scope, which motivates the development of our new benchmark. See §6 for broader related work.

| | Complex Procedure | High Dispersion | >8K Input | Real-world Tasks | >1K Output | Deterministic Solutions |
|---|---|---|---|---|---|---|
| NIAH (Kamradt, 2023) | ✗ | ✗ | ✓ | ✗ | ✗ | ✓ |
| RULER (Hsieh et al., 2024) | ✗ | ✗ | ✓ | ✗ | ✗ | ✓ |
| ZeroScrolls (Shaham et al., 2023) | ✗ | ✓* | ✓* | ✓* | ✗ | ✓* |
| ∞Bench (Zhang et al., 2024) | ✗ | ✗ | ✓ | ✓* | ✗ | ✓* |
| SumHaystack (Laban et al., 2024) | ✗ | ✗ | ✓ | ✓ | ✗ | ✗ |
| HELMET (Yen et al., 2025) | ✗ | ✓* | ✓ | ✓* | ✗ | ✓* |
| LongGenBench$_1$ (Liu et al., 2024) | ✗ | ✓ | ✓ | ✗ | ✓ | ✓ |
| LongGenBench$_2$ (Wu et al., 2024) | ✗ | ✗ | ✗ | ✗ | ✓ | ✓ |
| LongWriter (Bai et al., 2024) | ✗ | ✗ | ✗ | ✓ | ✓ | ✗ |
| LONGPROC (Ours) | ✓ | ✓ | ✓* | ✓* | ✓ | ✓ |

Table 1: Recent representative benchmarks for evaluating LCLMs. To the best of our knowledge, no previous benchmark encompasses all of the listed qualities. ✓*: the quality is featured by a subset of the tasks in a benchmark.

**Benchmarks focusing on long inputs.** The majority of existing benchmarks focus on long-context recall challenges (the first block in Table 1). The needle-in-the-haystack test (Kamradt, 2023, NIAH), one of the most widely used benchmarks, requires models to retrieve a target statement (needle) embedded within irrelevant text (haystack). Several subsequent benchmarks have extended this paradigm by incorporating multiple needles (Hsieh et al., 2024; Li et al., 2024a; Laban et al., 2024), introducing multi-step (but limited to a few steps of) reasoning (Levy et al., 2024; Li et al., 2024a), or employing more contextually relevant content (Liu et al., 2023; Karpinska et al., 2024; Wang et al., 2024b; Vodrahalli et al., 2024). While these extensions add complexity to the original NIAH framework, these dataset variants still exhibit relatively low dispersion (Goldman et al., 2024). While some benchmarks incorporate more realistic tasks (e.g., summarization and many-shot in-context learning) requiring more than a few snippets of relevant contexts (Shaham et al., 2023; Zhang et al., 2024; Yen et al., 2025), they only involve short output lengths (typically less than 100 words).

**Benchmarks focusing on long outputs.** Recent efforts have started exploring benchmarks requiring longer outputs (the second block in Table 1). Liu et al. (2024); Xu et al. (2024b) concatenate multiple problems to test LCLMs' ability to solve multiple tasks in a single pass. Wu et al. (2024) evaluate LLMs' capacity to generate repetitive information (e.g., year-long diary entries) under range-related or periodic constraints. However, these tasks concatenate independent problems without meaningful logical dependencies and segments in the required outputs are often disjointed. Bai et al. (2024) evaluate LCLMs through long-form content generation (e.g., creating a 30,000-word article on Roman Empire history), but such open-ended tasks make evaluation inherently subjective. Furthermore, none of these existing benchmarks adequately examine models' capabilities in multi-step reasoning and procedure following.

| | 0.5K Level | | | 2K Level | | | 8K Level | | | Access Info | Reasoning | Exec Search |
|---|---|---|---|---|---|---|---|---|---|---|---|---|
| | N | # In | # Out | N | # In | # Out | N | # In | # Out | | | |
| HTML to TSV | 100 | 12K | 0.5K | 189 | 23K | 1.3K | 120 | 38K | 3.7K | $\sqrt{}$(sequential) | – | – |
| Pseudocode to Code | 100 | 0.4K | 0.3K | 100 | 0.9K | 0.7K | – | – | – | $\sqrt{}$(sequential) | – | – |
| Path Traversal | 100 | 1.2K | 0.5K | 100 | 4.8K | 2.0K | 100 | 12K | 5.8K | $\sqrt{}$(targeted) | – | – |
| ToM Tracking | 100 | 2.0K | 0.5K | 100 | 2.5k | 2.0K | 100 | 4.1K | 7.9K | $\sqrt{}$(sequential) | $\sqrt{}$ | – |
| Countdown | 100 | 5.6K | 0.5K | 100 | 5.6K | 1.7K | 100 | 5.6K | 6.5K | – | $\sqrt{}$ | $\sqrt{}$ |
| Travel Planning | – | – | – | 100 | 6.0K | 1.2K | 100 | 6.0K | 5.3K | $\sqrt{}$(targeted) | $\sqrt{}$ | $\sqrt{}$ |

Table 2: Summary of tasks in LONGPROC. On the left, we show general statistics, including number of instances (N), and the average number of input and output tokens (# In/Out; counted with Llama-3 tokenizer). On the right, we compare tasks across three aspects on the requirements for accessing information, deductive reasoning, and executing search.

# 3 LONGPROC Benchmark

Recall that LONGPROC includes six diverse tasks. We provide task examples in Figure 1 and summarize the characteristics of these tasks in Table 2. In this section, we begin by describing the shared feature of **procedural generation** that underlies all tasks in LONGPROC. We then introduce each task in detail (§3.1) and analyze its distinct characteristics to highlight the diverse challenges they present for LCLMs (§3.2). Lastly, we explain the reliable evaluation metrics for these tasks (§3.3).

**Procedural generation.** The six tasks in LONGPROC share a common feature: each task requires LCLMs to execute a procedure to generate the output. Let $\Sigma$ denote the vocabulary. Given an input $\mathbf{X} \in \Sigma^*$, a procedure $\pi$ generates a **gold** output $\mathbf{Y}^* \in \Sigma^*$. The gold output $\mathbf{Y}^*$ is composed of a sequence of entries $\mathbf{Y}^* = \{y_1^*, y_2^*, ..., y_n^*\}$, where each $y_i^*$ is a structured form (e.g., text following a specific template such as a TSV row). The procedure $\pi$ is deterministic: at step $i$, exactly one correct entry $y_i^*$ exists, determined by $\pi$ based on the task input and all previous entries, i.e., $y_i^* = \pi(\mathbf{X}, y_1^*, y_2^*, ..., y_{i-1}^*)$.

We evaluate an LCLM by prompting it to execute an instruction $\mathbf{I} \in \Sigma^*$ (describing $\pi$) over $\mathbf{X}$: $\mathbf{Y} = \text{LCLM}(\mathbf{I}, \mathbf{X})$. To clearly specify a procedure $\pi$, the instructions contain both detailed step-by-step descriptions about the procedure (see Figure 2 for a concrete example) and few-shot examples. Since $\pi$ is deterministic, we can reliably evaluate the correctness of model actual prediction $\mathbf{Y}$ by comparing each predicted entry $y_i \in \mathbf{Y}$ against its corresponding gold entry $y_i^* \in \mathbf{Y}^*$ using **rule-based metrics**.

## 3.1 Tasks

We now introduce each of the tasks in LONGPROC with a focus on essential task characteristics. We discuss more detailed construction process in Appendix B and include **concrete examples for all tasks in Appendix F**.

**HTML to TSV.** This task (see top left of Figure 1 as well as Example F.1) requires LCLMs to extract query-specified information from HTML documents and organize the information into a table format (TSV). For instance, given an IMDB search result page in HTML format (with HTML tags) and a query specifying "extract the following properties from the items listed on the webpage: (1) Title; (2) Year; (3) Genre; (4) Rating", LCLMs are required to extract the data and format them into a table. We source the websites from Arborist (Li et al., 2024c) and manually annotate the questions and the ground truth TSV for the websites.

For this task, each entry $y_i$ corresponds to the step of processing the $i$-th item from the HTML document and format it into a TSV row. The primary challenge of this task is to robustly extract all relevant information from HTML and format them correctly.

**Pseudocode to Code.** This task, introduced in SPoC (Kulal et al., 2019), requires translating pseudocode into C++ code. See Example F.2 for a concrete example. The pseudocode is structured line-by-line, with each line corresponding directly to a line of C++ code, maintaining a one-to-one mapping between source and target.

Similarly to the HTML to TSV task, each entry $y_i$ represents the processing of the $i$-th line of pseudocode in the input, while the processing performs translation from pseudocode to C++ code as opposed to merely bookkeeping.

**Path Traversal.** This task (see Example F.3 for a concrete example) requires LCLMs to keep track of a route between two nodes, represented by a set of cities, in a graph where each city has **exactly one** outgoing connection to another city (node). Given a description of city connections and a source-destination pair, LCLMs must output a step-by-step route. It is important to note that this task does **not** require searching over a graph, since by construction, we constrain each city to have one and only one outgoing city, and we guarantee there exists one unique path from the source city to the destination city.

An entry $y_i$ in this task corresponds to the step of visiting to one city along the route, where each step transits into the unique outgoing connection from the current city to another city. This task challenges LCLMs to correctly retrieve the the connected outgoing city from the input descriptions and format the step into a standardized format at each step.

**Theory-of-Mind Tracking.** Inspired by a series of theory-of-mind reasoning datasets (Le et al., 2019; Sclar et al., 2023; He et al., 2023; Sprague et al., 2024), we design this task which requires tracking a person' beliefs about object locations in a dynamic environment. Given a story involving a sequence of person and object placements, LCLMs are requested to determine a person's belief about a specific object's location while considering the person's limited perspective. See bottom left of Figure 1 for an example.

This tasks differs from the previous three tasks by its requirement for **deductive reasoning**. Specifically, determining whether an person's belief should be updated requires inferring whether the person can observe the object (i.e., whether they are in the same room). The entry $y_i$ at a step $i$ records the detailed reasoning process by tracking the person's location, the object's location, the visibility condition, and the resulting belief state.

**Countdown.** Countdown is a game requiring to reach a target number using a list of given numbers along with four arithmetic operations ($+$, $-$, $\times$, and $/$) (see Figure 2 for a simplified example and Example F.5 for a concrete one). We source the problems from Gandhi et al. (2024).

For countdown, we instruct LCLMs to perform a *depth-first-search procedure*, which tests their capabilities in terms of carrying out an exhaustive search robustly. The entry $y_i$ at step $i$ is a filled template recording the state (the current set of numbers) and the actions (choosing two numbers to apply an operation or backtrack to previous states) taken at the state.

**Travel Planning.** This task (see right of Figure 1 for an example) adapted from Zheng et al. (2024) requires LCLMs to generate a multi-city travel plan that satisfies various constraints including fixed schedules, city visit durations, and direct flight availability between cities.

We also instruct the models to perform a depth-first-search procedure for this task. Here, a state in the search procedure represents a partial travel plan up to a date. LCLMs

```
Example Search Procedure for Countdown.

[INSTRUCTION]
We will follow this search process:
- At each state, first choose two numbers from the number set.
- Next, try the four operations (+, −, ×, and /) to obtain the
new number and add the new number to the number set.
- Continue this process until we reach the target number.

[EXAMPLE PROBLEM]
Numbers: [40, 19, 23, 7]
Target: 29

[EXAMPLE PROCEDURE]
Current number set: [40, 19, 23, 7]
 |- Pick two numbers (40, 19) (numbers left: [23, 7])
   |- Try 40+19=59. Current number set: [59, 23, 7]
     |- Pick two numbers (59, 23) (numbers left: [7])
       |- Try 59+23=82. Current number set: [82, 7]
         |- Try 82+7=89. Evaluate 89!=29. Drop this branch.
         |- Try 82-7=75. Evaluate 75!=29. Drop this branch.
         |- Try 82*7=574. Evaluate 574!=29. Drop this branch.
         |- Try 82/7=11.7. Evaluate 11.7!=29. Drop this branch
       |- Try 59-23=36. Current number set: [36, 7].
         |- Try 36+7=43. Evaluate 43!=29. Drop this branch.
         |- Try 36-7=29. Evaluate 29==29. Target found!

[SOLUTION]
40+19=59, 59-23=36, 36-7=29
```

Figure 2: Illustration of search procedure for Countdown (simplified). We provide both detailed instruction and example solving trace in our prompts to LCLMs.

need to check various constraints and explore feasible arrangements at each step. The entry $y_i$ is also a filled template recording the state computation for each scheduling decision.

## 3.2 Task Difficulty and Diverse Challenges

**Three difficulty levels.**   To evaluate models with different capabilities, we construct three difficulty levels in LONGPROC by selecting subsets of data points that require approximately 500, 2K, and 8K output tokens respectively (counted using Llama-3's tokenizer). Please refer to Appendix B for more details on how we obtain data points of different output lengths. Pseudocode to Code omits the 8K token set due to limited program lengths in the source SPoC dataset; Travel Planning excludes the 0.5K token set as even the simplest data points require more output tokens. Table 2 provides statistics and comparisons across tasks.

**Diverse challenges.**   We also highlight that the six tasks in LONGPROC, while sharing a common procedural generation framework, exhibit diverse challenges. The right side of Table 2 characterizes their key differences across three aspects:

• **Accessing Information:** Tasks vary in how they access and process context information. Some tasks, such as HTML to TSV and theory-of-mind tracking, require sequential processing of information in the input. Others, like Path Traversal and Travel Planning, need targeted retrieval of specific information at each step (e.g., identifying available out-going connections in Path Traversal).

• **Deductive Reasoning:** Tasks differ in the reasoning process required for executing the procedure. Theory-of-mind Tracking, Countdown, and Travel Planning involve different deductive reasoning capabilities. For instance, Countdown tests arithmetic computation, while Travel Planning needs compatibility checks between various scheduling constraints.

• **Executing Search:** Countdown and Travel Planning implement search-based procedures. In these cases, LCLMs are required to explore multiple possible actions at a step, and the solving process involves backtracking. This creates more complex execution traces that must record exploration paths and backtracking decisions.

Because all these tasks can be solved through pre-defined procedures that are already provided to LCLMs in prompts (Figure 2), their difficulty emerges from the extended generation requirements, where LCLMs must utilize information and preserve logical consistency across long distances. Therefore, our task design allows LONGPROC to evaluate multiple capabilities that are **central to long-context and long-form generation**.

## 3.3 Reliable Evaluation

Being able to reliably evaluate long outputs is a key feature of LONGPROC. Unlike existing benchmarks that rely on n-gram overlap metrics (Zhang et al., 2024; Shaham et al., 2023; Stelmakh et al., 2022; Fan et al., 2019) or LLM-based evaluation (Malaviya et al., 2024; Bai et al., 2024), LONGPROC enables reliable rule-based evaluation since its outputs are typically structured and deterministic. We evaluate task outputs as follows (see Appendix C for additional details):

• **HTML to TSV:** We compute row-level F1 scores over the model output rows $\mathbf{Y} = y_1, y_2, ...$ and ground truth rows $\mathbf{Y}^* = y_1^*, y_2^*, ....$

• **Pseudocode to Code:** Following Kulal et al. (2019), we evaluate translated functions using the unit tests. A function is correct if and only if it passes all test cases. This execution-based evaluation accommodates minor variations in implementation (e.g., using `printf` or `cout`).

• **Path Traversal and ToM Tracking:** These tasks require complete traces in a predefined format for deterministic processes. We use exact match evaluation $\mathbf{Y} = \mathbf{Y}^*$.

• **Countdown and Travel Planning:** For these search-based tasks, we evaluate the final solutions using rule-based validators. For Countdown, we verify calculation correctness of equations and whether we achieve the target. For Travel Planning, we verify satisfaction of all specified constraints.

| | Context Size | Average Scores | | |
| --- | --- | --- | --- | --- |
| | | 0.5K | 2K | 8K |
| *Open-weight models with less than 15B parameters* | | | | |
| Llama-3.2-1B-Inst | 128K | 4.0 | 0.1 | 0.0 |
| Llama-3.2-3B-Inst | 128K | 13.5 | 4.5 | 0.1 |
| Llama-3.1-8B-Inst | 128K | 29.7 | 20.7 | 5.3 |
| Llama-3-8B-ProLong | 128K | 20.1 | 9.0 | 4.4 |
| Mistral-7B-Inst-v0.3 | 32K | 18.6 | 12.4 | 1.2 |
| Phi3-7B-128k-Inst | 128K | 19.6 | 11.5 | 1.1 |
| Phi3-14B-128k-Inst | 128K | 25.4 | 12.2 | 2.5 |
| Qwen2.5-3B-Inst | 128K | 27.3 | 5.7 | 1.3 |
| Qwen2.5-7B-Inst | 128K | 27.8 | 23.0 | 3.8 |
| R1-Distill-Qwen2.5-7B$^{\mathcal{R}}$ | 128K | 16.3 | 7.7 | 2.4 |
| R1-Distill-Llama-3-8B$^{\mathcal{R}}$ | 128K | **51.6** | **24.6** | **7.5** |
| *Open-weight models with 15-75B parameters* | | | | |
| AI21-Jamba-1.5-Mini (52B) | 128K | 19.0 | 9.0 | 1.1 |
| Qwen2.5-32B-Inst | 128K | 68.4 | 50.3 | 17.1 |
| Qwen2.5-72B-Inst | 128K | 68.7 | 46.4 | 19.5 |
| Llama-3.1-70B-Inst | 128K | 72.9 | 58.0 | 24.2 |
| Llama-3.3-70B-Inst | 128K | 77.6 | 57.5 | 24.9 |
| R1-Distill-Qwen2.5-32B$^{\mathcal{R}}$ | 128K | 80.6 | 60.9 | 22.4 |
| R1-Distill-Llama-3-70B$^{\mathcal{R}}$ | 128K | **83.7** | **70.4** | **33.3** |
| *Proprietary models* | | | | |
| Claude-3-5-sonnet-2410 | 200K | 78.4 | 57.5 | 22.0 |
| GPT-4o-mini-24-07 | 128K | 55.7 | 38.1 | 7.6 |
| GPT-4o-2024-08 | 128K | **94.8** | **83.4** | 38.1 |
| Gemini-1.5-flash-001 | 1,000K | 78.9 | 52.3 | 15.3 |
| Gemini-1.5-pro-001 | 2,000K | 89.2 | 79.4 | **54.0** |

Table 3: Average performance across tasks of different LCLMs on LONGPROC at three difficulty levels (0.5K, 2K, 8K). The reasoning models are labeled as $\mathcal{R}$. All models show performance degradation with increased output length. Even frontier models struggle with 8K-token procedural generation tasks.

We note that the output format requirements in LONGPROC do not constrain model performance. Our experiments show top models (e.g., GPT-4o, Gemini-1.5-Pro) achieve near-perfect performance on the easiest set of these tasks, consistently maintaining proper formatting (§ 4). Also, structured output generation (e.g., JSON, TSV) is an important capability for practical applications. We believe a truly capable model should be able to comply with user-specified output formats.

# 4 A Comprehensive Evaluation of LCLMs on LONGPROC

**Setup.** We evaluate 23 LCLMs on LONGPROC across different output length configurations. For frontier closed-sourced models, we include GPT-4o (Achiam et al., 2023), Claude 3.5 (Anthropic, 2024), and Gemini 1.5 (Gemini Team, 2023; 2024). For open-weight models, we test various model families across different parameter scales and architectures, including ProLong (Gao et al., 2024), Llama-3 (Llama-3 Team, 2024), Mistral-v0.3 (Jiang et al., 2023), Phi-3 (Abdin et al., 2024), Qwen-2.5 (Yang et al., 2024), and Jamba (Lieber et al., 2024). We also cover several recently released reasoning models such as R1-distilled models (Guo et al., 2025).

Recall that LONGPROC includes three difficulty levels based on generation lengths: 500, 2K, and 8K tokens. Since different tokenizers produce varying numbers of tokens when encoding the same output, we allow an additional 0.5K-1K token buffer in generation length for all models to accommodate variations across different tokenizers. We use greedy decoding for all models given the deterministic nature of procedural generation. For reasoning models, we allow up to 16K tokens for generation to accommodate the additional

**HTML to TSV**

| | 0.5K | 2K | 8K |
|---|---|---|---|
| Llama-3.1-8B-Inst | 43.4 | 29.0 | 23.4 |
| Qwen2.5-7B-Inst | 45.2 | 32.1 | 17.0 |
| Qwen2.5-32B-Inst | 74.5 | 46.9 | 25.4 |
| R1-Distill-Qwen2.5-32B | 78.2 | 53.9 | 38.8 |
| Llama-3.3-70B-Inst | 78.0 | 60.2 | 51.7 |
| R1-Distill-Llama3-70B | 74.8 | 60.2 | 46.4 |
| GPT-4o-2024-08 | 87.0 | 76.4 | 65.5 |
| Gemini-1.5-pro-001 | 81.3 | 75.3 | 70.0 |

**Pseudocode to Code**

| | 0.5K | 2K | 8K |
|---|---|---|---|
| Llama-3.1-8B-Inst | 63.0 | 28.0 | - |
| Qwen2.5-7B-Inst | 60.0 | 31.0 | - |
| Qwen2.5-32B-Inst | 81.0 | 65.0 | - |
| R1-Distill-Qwen2.5-32B | 70.0 | 59.0 | - |
| Llama-3.3-70B-Inst | 76.0 | 64.0 | - |
| R1-Distill-Llama3-70B | 74.0 | 57.0 | - |
| GPT-4o-2024-08 | 90.0 | 84.0 | - |
| Gemini-1.5-pro-001 | 81.6 | 50.0 | - |

**Path Traversal**

| | 0.5K | 2K | 8K |
|---|---|---|---|
| Llama-3.1-8B-Inst | 17.0 | 0.0 | 0.0 |
| Qwen2.5-7B-Inst | 0.0 | 0.0 | 0.0 |
| Qwen2.5-32B-Inst | 29.0 | 0.0 | 0.0 |
| R1-Distill-Qwen2.5-32B | 97.0 | 61.0 | 0.0 |
| Llama-3.3-70B-Inst | 70.0 | 1.0 | 0.0 |
| R1-Distill-Llama3-70B | 95.0 | 90.0 | 31.0 |
| GPT-4o-2024-08 | 98.0 | 77.0 | 34.0 |
| Gemini-1.5-pro-001 | 97.0 | 96.0 | 81.0 |

**ToM Tracking**

| | 0.5K | 2K | 8K |
|---|---|---|---|
| Llama-3.1-8B-Inst | 17.0 | 0.0 | 0.0 |
| Qwen2.5-7B-Inst | 2.0 | 0.0 | 0.0 |
| Qwen2.5-32B-Inst | 65.0 | 8.0 | 0.0 |
| R1-Distill-Qwen2.5-32B | 67.0 | 50.0 | 0.0 |
| Llama-3.3-70B-Inst | 87.0 | 45.0 | 0.0 |
| R1-Distill-Llama3-70B | 76.0 | 59.0 | 7.0 |
| GPT-4o-2024-08 | 100.0 | 77.0 | 0.0 |
| Gemini-1.5-pro-001 | 92.0 | 71.0 | 28.0 |

**Countdown**

| | 0.5K | 2K | 8K |
|---|---|---|---|
| Llama-3.1-8B-Inst | 8.0 | 12.0 | 3.0 |
| Qwen2.5-7B-Inst | 32.0 | 36.0 | 2.0 |
| Qwen2.5-32B-Inst | 96.0 | 87.0 | 55.0 |
| R1-Distill-Qwen2.5-32B | 91.0 | 88.0 | 51.0 |
| Llama-3.3-70B-Inst | 77.0 | 89.0 | 61.0 |
| R1-Distill-Llama3-70B | 99.0 | 86.0 | 47.0 |
| GPT-4o-2024-08 | 99.0 | 95.0 | 67.0 |
| Gemini-1.5-pro-001 | 94.0 | 84.0 | 46.0 |

**Travel Planning**

| | 0.5K | 2K | 8K |
|---|---|---|---|
| Llama-3.1-8B-Inst | - | 55.0 | 0.0 |
| Qwen2.5-7B-Inst | - | 39.0 | 0.0 |
| Qwen2.5-32B-Inst | - | 95.0 | 5.0 |
| R1-Distill-Qwen2.5-32B | - | 54.0 | 22.0 |
| Llama-3.3-70B-Inst | - | 86.0 | 12.0 |
| R1-Distill-Llama3-70B | - | 71.0 | 35.0 |
| GPT-4o-2024-08 | - | 91.0 | 24.0 |
| Gemini-1.5-pro-001 | - | 100.0 | 45.0 |

Figure 3: Performance comparison across tasks and output lengths (grey blocks indicate unavailable configurations).

"thinking" tokens. Otherwise, they often fail to complete generation. In contrast, instruction-tuned models do not effectively utilize additional tokens, as they typically terminate early even when given a higher token limit. We provide in-context examples along with solving procedures in prompts for all tasks, except for HTML to TSV and Pseudocode to Code (see Appendix H for detailed prompts).

**Results.** Table 3 summarizes LCLM **average performance** across tasks at different lengths.

**Existing models struggle in extensive long procedural generation.** Frontier proprietary models demonstrate the best performance. GPT-4o and Gemini-Pro achieve near-perfect scores on 0.5K tasks and maintain strong performance at 2K. However, they experience substantial performance degradation at 8K, falling well short of their claimed context windows of 128K tokens or more. Open-weight models lag significantly behind proprietary models. Models under 15B parameters struggle with 2K tasks, with the best performer, R1-Distill-Llama-3-8B, reaching only 24.6. Mid-sized models (13B–70B parameters) handle some 8K tasks but achieve substantially lower performance than frontier models.

**Model scale is critical for task performance.** We observe substantial performance differences across scales within model families, as evidenced by the gaps between Llama3.1-8B versus Llama3.1-70B and Qwen-8B versus Qwen-72B. While differences caused by model families are less pronounced than scale-related gaps, notable performance variations still exist between similarly-sized models. For example, Llama-3.1-70B outperforms Qwen2.5-72B by approximately 30% (relative gain) on both 2K and 8K token sets.

**Reasoning models outperform their instruction-tuned counterparts.** For example, R1-Distill-Qwen2.5-32B outperforms Qwen2.5-32B-Inst by approximately 10% relative gain across all three difficulty levels. This performance gap suggests a potential synergy between long CoT training and long-form generation capabilities. We provide a more detailed analysis of reasoning models in §5.

# 5 Analysis

Our analysis focuses on the challenges of long procedural generation and discussion around reasoning models. Refer to Appendix A for additional analysis, such as comparison to low dispersion task (RULER), small scale human evaluation, and qualitative analysis

**Comparison of performance across tasks.** Figure 3 presents the performance comparison across tasks and generation lengths for 8 representative models. We leave the complete results of all models in Appendix E. In general, We observe **steeper performance degra-**

**dation in tasks requiring long-range reasoning**. Gemini-1.5-Pro, the best model on the 8K set overall, shows more significant performance decline on tasks requiring reasoning (ToM Tracking, Countdown, and Travel Planning) compared to more straightforward tasks (HTML to TSV). This discrepancy arises from the dependent nature of reasoning tasks, where generating each entry relies on context from previous entries. Such variances in performance degradation rate also highlight the task diversity in LONGPROC.

**Models struggle to maintain long-range coherence.** We analyze how models degrade during generation to better understand the challenges of long procedural generation. Specifically, we examine output correctness across different segments of the generated text. We divide $\mathbf{Y}$ into four even segments $\mathbf{Y}^{*(1)}$, $\mathbf{Y}^{*(2)}$, $\mathbf{Y}^{*(3)}$, and $\mathbf{Y}^{*(4)}$, and evaluate the model correctness for each segment. When evaluating the correctness of the $i$-th segment, we *prefill the prompt with ground truth segments* $\mathbf{Y}^{*(1)}$ to $\mathbf{Y}^{*(i-1)}$ to make generation conditioned on the correct prefix.

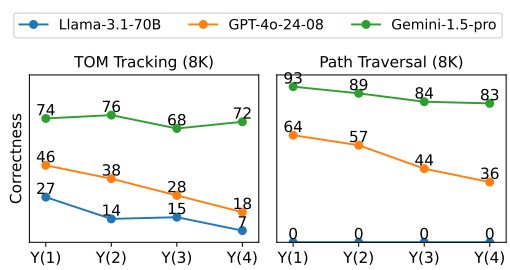

Figure 4: Performance of different segments in the outputs. Models achieve lower performance in the later segments.

Figure 4 shows the per-segment correctness on Path Traversal and ToM Tracking in the 8K token setting. We observe clear **performance degradation in the later segments of the generation**. This trend indicates that, as the generated outputs accumulate, models increasingly struggle to maintain coherence with both the input and previously generated text. We also provide a qualitative analysis in Appendix A.4, which reveals that LCLMs make both recall errors and reasoning errors when handling long-range dependencies.

**Comparison between reasoning models and instruct models across tasks.** Results in §4 suggest that the reasoning models achieve strong overall performance, and we further investigate which specific tasks benefit more from long CoT training. As shown in Figure 3, the most substantial improvements are observed in Path Traversal, with R1-Qwen-32B also showing gains in HTML to TSV compared to Qwen-32B-Inst. Interestingly, these two tasks are considered less reasoning-intensive, with their difficulty mainly lying in identifying relevant information amid similar contexts. For the three reasoning-focused tasks (bottom row), reasoning models substantially outperform their instruction-tuned counterparts in ToM Tracking and Travel Planning (8K). However, reasoning models show some performance degradation on Countdown (8K) and Travel Planning (2K). This occurs because they cannot generate final solutions within the 16K output token budget (see Appendix A.2 for details).

**Benefits of procedural generation.** We highlight that the ability to execute long-form procedures is advantageous for applications requiring extended reasoning steps. In Figure 5, we compare models' performance on two tasks requiring systematic search procedures when prompted using two methods: 1) ICL (in-context learning) that provides input and output pairs (where output examples also help specify the formats) 2) ICL with Procedure that provides detailed procedure (see Figure 2 for a concrete example).

As shown in Figure 5, using procedures in prompt generally improves performance for both instruct and reasoning models. No-

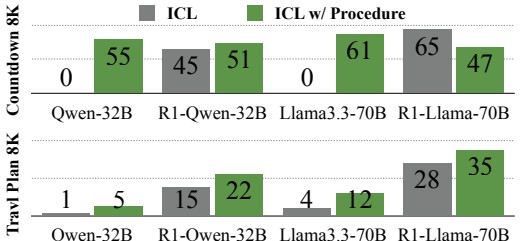

Figure 5: Performance comparison between standard ICL (input-output pairs) and ICL with procedure. Using procedure in prompts can enhance performance of both instruct and reasoning models.

tably, instruct models prompted with procedures achieve substantial performance gains compared to ICL, sometimes even reaching comparable performance to reasoning models with standard ICL. We note that even with standard ICL, models are still instructed to "think carefully" and allowed to use CoT. Our findings suggest that instruct models already possess branching and backtracking capabilities, which are often characterized as distinctive features of reasoning models (Guo et al., 2025).

## 6    Related Work

**Improving long-context modeling.**   Language models' context sizes have been significantly expanded thanks to recent advances in pre-training and post-training methods (Gao et al., 2024; Fu et al., 2024; Xiong et al., 2023; An et al., 2024; Hu et al., 2024), alternative attention mechanisms (Xiao et al., 2023; Bertsch et al., 2023; Jin et al., 2024; Yen et al., 2024) or architectures (Gu & Dao, 2024; Lieber et al., 2024; Peng et al., 2023), context compression approaches (Xiao et al., 2023; Xu et al., 2024a; Zhang et al., 2023), and position extrapolation techniques (Chen et al., 2024; 2023; Ding et al., 2024; Peng et al., 2024; Zhu et al., 2024). While we primarily focus on evaluating different model families, our dataset can also be used to assess the effectiveness of position extrapolation or context compression techniques.

**Evaluating long-context models.**   In §2, we have discussed the limitations of multiple commonly used existing LCLMs benchmarks. In addition to those discussed in §2, some application-centric benchmarks focus on specific domains (Wang et al., 2024b; Karpinska et al., 2024; Chang et al., 2024; Kim et al., 2024; Wang et al., 2024a; Li et al., 2024b; Bertsch et al., 2024; Lee et al., 2024). In contrast, LONGPROC focuses on general long-form generation, covering a diverse set of tasks while requiring significantly longer generation lengths.

**Executing procedures with models.**   There also exists prior work that studies models' performance on specific operations like addition to investigate their length generation capabilities (Nye et al., 2021; Dziri et al., 2023; Zhou et al., 2024) or algorithmic reasoning capabilities (Gu et al., 2024; Wang et al., 2023; Gandhi et al., 2024; Valmeekam et al., 2023; Sel et al., 2024; Borazjanizadeh et al., 2024). Our benchmark features a broader range of procedures beyond algorithmic ones and emphasizes extended length to test long-form generation capabilities. In particular, DOLOMITES (Malaviya et al., 2024) evaluates models on methodical writing tasks given general procedural steps (high-level plans as opposed to detailed procedures). However, its inputs and outputs are short by the standard of LCLMs and its solutions are open-ended solutions.

## 7    Conclusion

In this work, we introduced LONGPROC, a benchmark designed to evaluate LCLMs' capability for long procedural generation tasks. Our evaluation across 23 LCLMs reveals significant challenges in generating coherent, lengthy procedural content, even for state-of-the-art models. While larger models show greater resilience to increasing generation lengths, all models struggle with robust generation at the 8K token level, even if they achieve strong performance on current commonly used long-context recall benchmarks (such as RULER). Our findings highlight the limitations of current LCLMs in handling complex, long-form procedural generation tasks and suggest substantial room for improvements.

## Acknowledgments

We acknowledge members of Princeton Language and Intelligence for their helpful feedback and discussion. This work is gratefully supported by NSF CAREER award (IIS-2239290), NSF CAREER award (IIS-2145280), the NSF AI Institute for Foundations of Machine Learning (IFML), and a grant from Intel. Howard Yen is supported by the William A. Dippel' 50 * 55 Graduate Fellowship.

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

# A Additional Analysis

## A.1 Comparison with Low Dispersion Tasks

We compare models' performance on LONGPROC against their performance on RULER (Hsieh et al., 2024), a commonly used long-context recall benchmark with relatively low dispersion. Table 4 shows the comparison on LONGPROC and RULER. We mark the total lengths (input plus output) under each task and include one model from each of the three model groups.

|  | HTML 42K | Path 17K | ToM 12K | RULER 64K | 128K |
|---|---|---|---|---|---|
| Llama-3.1-8B | 23.4 | 0.0 | 0.0 | 89.8 | 81.3 |
| Llama-3.1-70B | 47.0 | 0.0 | 0.0 | 90.5 | 75.8 |
| GPT-4o-24-08 | 75.4 | 34.0 | 0.0 | 95.6 | 91.3 |

Table 4: Performance comparison between LONGPROC and RULER. We show **overall text lengths (input + output)** under the task.

The results suggest **substantial performance gaps between high-dispersion tasks in LONGPROC and low-dispersion tasks in RULER**. While models across all scales (even those below 10B parameters) achieve strong performance on RULER at 64K tokens and some even maintain strong recall capabilities up to 128K tokens, none demonstrate robust generation capabilities on LONGPROC with much shorter overall lengths. This contrast highlights the challenge of processing diffused information, aligning with recent calls for developing truly challenging long-context benchmarks that incorporate high information dispersion (Goldman et al., 2024).

## A.2 Additional Details on the Comparison Between Reasoning Models and Instruct Models

|  | Countdown (2K) | | | Countdown (8K) | | | Travel Plan (2K) | | | Travel Plan (8K) | | |
|---|---|---|---|---|---|---|---|---|---|---|---|---|
|  | acc | term | acc/term | acc | term | acc/term | acc | term | acc/term | acc | term | acc/term |
| Qwen2.5-32B-Inst | 87 | 88 | 99 | 55 | 78 | 71 | 95 | 99 | 96 | 5 | 82 | 6 |
| R1-Distill-Qwen-32B | 88 | 88 | 100 | 51 | 52 | 98 | 54 | 56 | 96 | 22 | 36 | 61 |
| Llama-3.1-70B-Inst | 89 | 91 | 98 | 61 | 68 | 90 | 86 | 94 | 92 | 12 | 60 | 20 |
| R1-Distill-Llama-70B | 86 | 86 | 100 | 47 | 51 | 92 | 71 | 77 | 92 | 35 | 64 | 55 |

Table 5: Comparison of accuracy (acc), termination rate (term; whether model complete generation and output EOS tokens within the allocated budget), and accuracy among terminated generations (acc/term). Reasoning models underperform compared to instruct models mainly because they fail to terminate within constraints, thus not providing final solutions.

In §5, we compare the performance of reasoning models and instruct models across tasks. Recall that we observe reasoning models underperform their instruct counterparts on Travel Planning (2K) and Countdown (8K). This poor performance is due to the fact that reasoning models cannot finish generation and produce final solutions within a 16K output token budget, due to their use of a separate "thinking" stage. Note that instruct models are given a budget of 4K for Travel Planning (2K) and a budget of 10K for Countdown (8K), significantly lower than the budget of reasoning models (16K for all settings).

Table 5 shows models' accuracy (acc), termination rate (term, which measures whether models complete generation within the allocated token budget), and accuracy among terminated generations (acc/term). On Countdown (8K) and Travel Planning (2K), where reasoning models underperform their instruct counterparts, reasoning models actually achieve the same or higher accuracy among terminated runs. However, their substantially lower termination rates lead to inferior overall accuracy. On Travel Planning (8K), reasoning models demonstrate much higher accuracy among terminated generations. Nevertheless, considering that reasoning models are allocated significantly more tokens yet achieve lower termination rates on Countdown (8K) and Travel Planning (2K), this reveals the token-inefficiency of reasoning models for certain problems, opening an interesting research direction for future work.

### A.3 Human Evaluation

|  | Countdown | Travel |
|---|---|---|
| GPT-4o | 68.6 | 14.2 |
| Gemini-pro | 32.3 | 40.0 |
| Human | **100.0** | **91.4** |

Table 6: Human evaluation results on Countdown and Travel Planning that suggest substantial gaps between best LCLM performance and human performance.

To validate model performance against human capabilities, we conducted a small-scale human evaluation study. Our authors, who **did not** participate in the corresponding generation process of Countdown and Travel Planning, manually attempted 35 8K-level problems from each of these two tasks. We leverage the same prompt and evaluation setup as in our model evaluation. As shown in Table 6, our human evaluators achieves an accuracy of 100% on Countdown and 91% on Travel Planning, while frontier LCLMs only solved around 68% and 40% of the same problems, respectively. This comparison demonstrates a substantial performance gap between current LCLMs and human capability.

### A.4 Qualitative Analysis

|  | **Main Capability** | Typical Recall Errors | Typical Reasoning Errors |
|---|---|---|---|
| HTML to TSV (8K) | Extract Info | skipping rows (8/20) hallucinating details (7/20) | – |
| ToM Tracking (8K) | Reasoning | – | incorrect inferences about locations changes (19/20) |
| Travel Planning (8K) | Search | hallucinating non-existential direct flights (10/20) | incorrect state updates and state transitions (8/20) |

Table 7: Summary of qualitatively analysis on outputs (GPT-4o). GPT-4o makes both long-context recall and long-range inference errors in the extensive generation process. Note that GPT-4o rarely makes these errors in easier settings (0.5K or 2K) as demonstrated by its almost perfect performance.

We qualitatively analyze the typical errors made by LCLMs. Table 7 provides a summary of these errors, with concrete examples available in Appendix G. We broadly categorize these errors into two types: long-context recall errors (where the model restates information inconsistent with the context) and long-range reasoning errors (where the model outputs entries that cannot be logically deduced from the context and previous entries). We find that the frequency of both error types increases substantially as task difficulty increases. In contrast, GPT-4o rarely exhibits these errors in the 0.5K or 2K settings, where it achieves almost perfect performance.

**Errors in extracting information.** On the HTML to TSV task, models only need to copy information from the input. We analyzed 20 errors at 8K tokens made by GPT-4o on the HTML to TSV task. We found that 8 out of 20 errors were caused by the model skipping intermediate rows or only generating the first few rows, and 7 errors were caused by the model hallucinating details in some properties (e.g. adding a wrong street name to the address property). These errors indicate the model is not able to consistently identify and recall all relevant information from long context windows to the long outputs.

**Errors in deductive reasoning.** In the ToM Tracking task, models must perform long-range deduction by inferring location changes of people or objects based on distant context. We analyzed 20 errors made by GPT-4o. Our analysis reveals that 19 out of 20 errors in this task stemmed from incorrect long-range inferences about location changes, rather than reasoning within local context windows (e.g., determining if a person and object are in the same room).

**Errors in executing search.** In search-based tasks, models struggle with both following complex search procedures and maintaining context adherence. Analysis of 20 GPT-4o errors in Travel Planning shows that 10 cases involved hallucinating non-existent direct flights between cities, while 8 cases demonstrated failures in either exploring possible options or correctly updating search states (Appendix G).

## A.5 Analysis of Performance Degradation Pattern

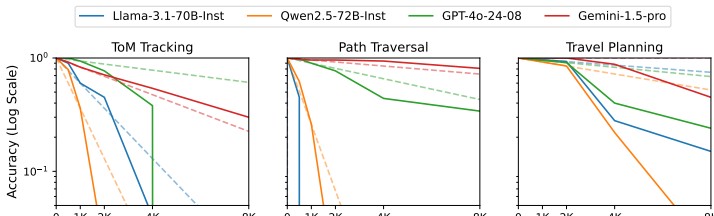

Figure 6: Analysis of performance with respect to generation length. Dashed lines represent *hypothetical* trends assuming constant per-entry error rates (where log accuracy is proportional to length). While this model fits certain cases, such as Gemini-1.5-Pro on ToM tracking and Path Traversal, the observed performance trends generally indicate higher error rates at longer lengths, suggesting models struggle to maintain coherence over extended sequences.

We analyze whether performance degradation follows a **simple error accumulation** pattern as generation length increases. Under an error accumulation assumption, if LCLMs have a probability $p$ of correctly predicting each entry $y_i$, then for a procedure of length $L$ with average entry length $t$, the overall accuracy should be $p^{\frac{L}{t}}$. This implies the log-accuracy should be proportional to length: $\log(acc) \propto L$.

Figure 6 compares actual log-scaled model accuracy against hypothetical trends extrapolated from model performance at 1K and 2K tokens. We find that the observed performance degradation generally deviates from hypothetical trends, except for Gemini-1.5-Pro on ToM tracking and Path Traversal tasks. The increasing per-entry error rate suggests that *performance deterioration extends beyond simple error accumulation*, highlighting the challenges in long-form generation.

## A.6 Analysis on Robustness to Irrelevant Context

|  | Extract All (No Filter) | Filtering |
| --- | --- | --- |
| LLaMA-3.1-70B | 55.7 | 38.4 |
| GPT-4o-2024-08 | 78.1 | 52.8 |
| Gemini-1.5-Pro-001 | 77.5 | 62.5 |

Table 8: Performance on HTML-TO-TSV (8K) task with and without filtering. The filtering setting introduces more irrelevant information, leading to a notable drop in performance.

While our primary focus is on long-form generation and synthesis over dispersed information, we include an analysis on model robustness to irrelevant context through the HTML to TSV task. This task includes two settings (refer to Appendix B.1 for details on how this task is constructed): 1) **Extract All (No Filter):** The model is asked to extract all relevant items from the input HTML page (e.g., "all movies in the page"), and 2) **Filtering:** The model must extract only a subset of relevant items that satisfy certain conditions (e.g., "all action movies in the page").

The filtering setting introduces additional irrelevant information into the context and requires the model to robustly identify and extract only the relevant items. Table 8 shows the performance of several representative models on both settings for the 8K-token output level. We observe a substantial drop in the filtering setting, suggesting that current LLMs struggle with selective extraction in the presence of irrelevant information.

# B Details of Dataset Generation

## B.1 HTML to TSV

This task involves extracting the information from an HTML page into a structured tab-separated TSV output. An example can be found in Example F.1.

We build upon the *arborist* dataset from Li et al. (2024c) where each data example includes a set of web pages of a website scraped from the Internet and a task a program synthesis model can perform based on the web pages. We manually inspect the arborist dataset and select 56 websites that satisfy the following conditions: (1) The web pages contain at least one structured table-like component that can be converted into TSV files; (2) The table-like component has enough rows and properties so that the corresponding TSV file has more than 2K tokens.

For each of the selected websites, we perform the following cleaning steps: (1) If the website consists of multiple HTML files, we merge them into a single HTML file from which the table component can be extracted; (2) We remove any CSS style, metadata, JavaScript script, and comment from the HTML file, and remove unnecessary attributes of HTML tags (such as alt text); (3) We simplify the cleaned HTML tree by removing empty tags; (4) We manually check the cleaned HTML file to make sure that a table can still be extracted from it.

After cleaning the input HTML file, we first extract the ground truth TSV file using heuristics (e.g. extracting the table tag from the HTML file). Because natural websites on the Internet do not necessarily use the formats from common heuristics, we manually annotate the ground truth TSV when the heuristics fail.

We compute the length (in tokens) of the extracted ground truth TSV files, and depending on the length of the ground truth of each website, we split each website into non-overlapping subsamples so that we can obtain more data points for the 8K level and can create the 2K and 0.5K levels. If the ground truth TSV table exceeds 10K tokens, we split the table in half into two non-overlapping tables, each with 2K-8K tokens. After splitting the ground truth, we split the cleaned input HTML file into two HTML files accordingly. In this way, we obtain two data points for the 8K test set by subsampling from a single website. Similarly, we split each website into 1-3 subsamples at the 2K output level and 1-3 subsamples at the 0.5K output level.

To make the task more challenging, we add a filtered version of each website at the 8K and 2K levels where the model is prompted to extract only some rows based on a given specification. Note that the output length after filtering is shorter than the length without filters. We manually set the filtering condition such that the output length of an 8K-level input after filtering is at least 2K, and the length of a 2K level input after filtering is at least 0.5K.

## B.2 Pseudocode to Code

This task involves translating pseudocode into C++ code, with a one-to-one correspondence between pseudocode and C++ lines. For examples of this translation, see Figure F.2, and for the evaluation prompt, see Prompt H.2.

We use the dataset from the original SPOC paper (Kulal et al., 2019), where each example includes line-by-line pseudocode annotations and associated test cases. A C++ translation is considered correct only if it passes all test cases. To create our test sets, we randomly sample programs from the combined training, development, and test splits of the original dataset. We formed two test sets: one containing 200 randomly sampled programs with outputs less than 0.5K tokens, and another 200 randomly sampled programs with outputs greater than 0.5K tokens.

### B.3 Path Traversal

This tasks requires LCLMs to keep track of a route between two cities in a hypothetical transit network between cities. Please refer to Prompt H.3 for a concrete data example and the detailed prompt used for evaluation.

Essentially, the underlying problem is to traverse a path between two nodes (cities) in a directly acyclic graph. Each data instance contains a graph $G = \langle V, E \rangle$, where $V$ represents the set of nodes and $E$ represents the set of directed edges. The task asks LLMs to find the path between a source node $v_s$ and a destination node $v_d$.

We construct the graph and source-target node pairs with two important constraints: 1) there exists exactly one path from the source node $v_s$ to the target node $v_d$, and 2) each node $v_i$ along the path has exactly one outgoing edge to the next node $v_j$ along the path. This design ensures that LLMs can find the path by simply following the next node from the initial node, without requiring any search algorithms. To cast the problem into a natural language task, we assign each node a real city name from a pre-collected set and describe each edge using natural language. The edge descriptions follow this template: "`[city_a]` `is a lively city. You can travel from [city_a] to [city_b] by [transit].`" The complete list of edges is provided to LLMs as a natural language description of the graph. We construct the data sets for the three difficulty levels by varying the number of the cities in the output path.

### B.4 Theory-of-Mind Tracking

This task contains diverse theory-of-mind (ToM) story-question pairs that require LCLMs to track the locations and beliefs in stories about object placement. The stories follows the Sally-Anne style Baron-Cohen et al. (1985), with adaptions inspired by He et al. (2023). Each story attaches one first-order ToM question inquiring about an agent's belief of an object's location. See Prompt F.4 for an example data point.

The story consists of a certain number of **steps**, which controls the difficulty of the story-question pair. We include five types of steps: (1) an agent moves from one room to another, (2) an agent moves an object (and the agent themselves) from one room to another, (3) an agent moves an object from one container to another within the same room, (4) an agent leaves the current room, (5) an agent enters a room. Each step stems from a random selection among the five types.

The question has the form of *"Where does Agent believe Object is?"* where the target *Agent* and target *Object* are randomly chosen from the story. To reason about the agent's belief, one needs to track the steps in which the agent's belief changes, in particular, the steps where the agent sees the object being moved. As the target output in F.4 exemplifies, the search procedure mainly tracks four key aspects in each step: (1) location of target agent, (2) location of target object, (3) whether target agent sees target object, (4) target agent's current belief of target object's location. We instruct the models to closely follow this search procedure.

### B.5 Countdown

Recall that the Countdown game requires to reach a target number using basic arithmetic operations on a given list of numbers. We adapted the data generation scripts from Gandhi et al. (2024) for use in LONGPROC. Specifically, we only use four number games, and restrict the values of numbers to be less than or equal to 50. This is to emphasize the challenges in procedural execution instead of numerical computation.

For this task, we employ a depth-first-search (DFS) procedure following prior work (Yao et al., 2023; Gandhi et al., 2024). We detail the pseudocode for the underlying DFS algorithm of the procedure in Algorithm (1). In this search algorithm, each state represents the current set of numbers. The actions and state transitions involves choosing two numbers and an arithmetic operation to apply to them to transit to a new state. The search terminates when a leaf node matches the target number.

Since the search procedure ends whenever we hit the target number, this leads to varying number of output tokens in the final trace. To create our test sets of different difficulty levels, We randomly create a large pool of data points (10,000) and sample subsets according to the number output tokens to form our test set.

---

**Algorithm 1** DFS procedure for the Contdown task.

---

```
 1: function COUNTDOWNDFS(nums, target)
 2:     if len(nums) == 1 then
 3:         return nums[0] == target
 4:     end if
 5:     for a, b in CHOOSETWOELEMENTS(nums) do
 6:         remaining_nums ← nums - {a, b}
 7:         a, b ← max(a, b), min(a, b)
 8:         for op in [+, -, *, /] do
 9:             if COUNTDOWNDFS(remaining_nums + op(a, b), target) then
10:                 return True
11:             end if
12:         end for
13:     end for
14:     return False
15: end function
```

---

### B.6 Travel Planning

This task requires models to generate multi-city travel plans that satisfy various constraints. We adapt data from Zheng et al. (2024). While the original authors evaluated the task using few-shot prompting without detailed procedures, we explicitly provide the solution procedure in our prompts.

We also implement a depth-first search (DFS) procedure, where each state represents a partial travel plan up to a given day. At each step, LLMs must verify various constraints and explore feasible arrangements. The complete pseudocode for this DFS procedure is detailed in Algorithm 2.

The search terminates when a complete valid plan is found, resulting in solution traces of varying lengths. From the original dataset of 1,600 examples, we sampled subsets based on their output token lengths to create three test sets of different difficulty levels.

## C Details of Evaluation Metrics

Recall that We require LONGPROC uses rule-based metrics for reliable evaluation of model outputs. We require LCLMs to format their answers in specific formats. To enable compliance with these output formats, we include descriptions and examples of the formats in prompts (see Appendix H for examples). Additionally, we instruct models to mark their answers with designated tags (e.g., enclosing the final plan for Travel Planning between <plan> and </plan>). This approach allows models the flexibility to generate supplementary content, such as chain-of-thought reasoning, before providing their final answers.

For each task, we alsop implement specific normalization procedures to accommodate slight variations of styles and wording in the outputs. Specifically, we evaluate the outputs for each task as follows:

- **HTML to TSV:** We compute row-level F1 scores over the model output rows $\mathbf{Y} = y_1, y_2, ...$ and ground truth rows $\mathbf{Y}^* = y_1^*, y_2^*, ....$ A row is considered correct if every column in a row matches the ground truth. We apply standard normalization steps widely used in QA evaluation (lower casing, removing extra whitespaces and punctuations) to accommodate slight variations in answers.
- **Pseudocode to Code:** Following Kulal et al. (2019), we evaluate translated functions using the unit tests. A translation is correct if and only if it passes all test cases. This

---

**Algorithm 2** DFS procedure for the Travel Planning task.

---

```
 1: function TRAVELPLANDFS(current_day, current_schedule, remaining_cities, fixed_schedules)
 2:     if ISCOMPLETE(current_schedule) then
 3:         return current_schedule
 4:     end if
 5:     if current_day in fixed_schedules then
 6:         choices ← fixed_schedules[current_day]
 7:     else
 8:         choices ← remaining_cities
 9:     end if
10:     last_city ← TAIL(current_schedule)
11:     for chosen_city in choices do
12:         if not EXISTSDIRECTCONNECTION(last_city, chosen_city) then
13:             continue
14:         end if
15:         end_day ← current_day + chosen_city.duration
16:         if not COMPATIBLEWITHFIXEDSCHEDULE(end_day, fixed_schedules) then
17:             continue
18:         end if
19:         branch_result ← TRAVELPLANDFS(end_day, current_schedule + [chosen_city],
    remaining_cities - {chosen_city}, fixed_schedules)
20:         if branch_result is not None then
21:             return branch_result
22:         end if
23:     end for
24:     return None
25: end function
```

---

execution-based evaluation accommodates minor variations in implementation (e.g., using `printf` or `cout`).

- **Path Traversal and ToM Tracking:** These tasks require complete traces in a predefined format for deterministic processes. We use exact match evaluation $\mathbf{Y} = \mathbf{Y}^*$. We also apply these standard normalization steps before evaluating exact matches.

- **Countdown and Travel Planning:** For these search-based tasks, we evaluate the final solutions using rule-based validators. For Countdown, we verify calculation correctness of equations and whether we achieve the target. For Travel Planning, we verify satisfaction of all specified constraints. We use the final answer format and rule-based validators from the original implementations by Gandhi et al. (2024) and Zheng et al. (2024), respectively. The final outputs comply with the specified format in the final (short-form) solutions, as is the standard approach for evaluating correctness. We do not evaluate the search trace generated by the models, as these may exhibit greater variations.

We find that models are able to follow the output format effectively. Our experiments show that top models (e.g., GPT-4o, Gemini-1.5-Pro) achieve near-perfect performance on the easiest set of these tasks, consistently maintaining proper formatting (§ 4). When errors occur, they stem from content rather than formatting issues (see Appendix A.2 for analysis).

**Handling the thinking tokens for reasoning models** Reasoning models often invoke a "thinking" stage. In our evaluation, we preserve these thinking tokens rather than removing them. As mentioned earlier, we require LCLMs to mark formulated answers with special tags, allowing us to extract answers using these tags. We observe that reasoning models occasionally format answers during their thinking stages. Therefore, instead of stripping out thinking tokens, we use tags to extract answers, which improves the performance measurement of reasoning models.

## D   Compute Cost of Evaluation

For 10B-scale models, we used a single H100 80GB GPU for approximately 9 GPU hours per model. For 70B-scale models, we used 4×H100 GPUs with a total compute time of around 110 GPU hours.

# E  Detail Results for All LCLMs across Tasks

| Results on 0.5K set | | | | | | | |
| --- | --- | --- | --- | --- | --- | --- | --- |
| | HTML to TSV | Pseudocode to Code | Path Traversal | ToM Tracking | Countdown | Travel Planning | Average |
| Llama-3.2-1B-Inst | 4.1 | 6.0 | 0.0 | 0.0 | 10.0 | − | 4.0 |
| Llama-3.2-3B-Inst | 7.6 | 10.0 | 0.0 | 0.0 | 50.0 | − | 13.5 |
| Llama-3.1-8B-Inst | 43.4 | 63.0 | 17.0 | 17.0 | 8.0 | − | 29.7 |
| Llama-3-8B-ProLong-512k-Inst | 47.7 | 28.0 | 0.0 | 3.0 | 22.0 | − | 20.1 |
| Mistral-7B-Inst-v0.3 | 43.2 | 38.0 | 0.0 | 2.0 | 10.0 | − | 18.6 |
| Phi-3-small-128k-Inst | 26.9 | 54.0 | 0.0 | 7.0 | 10.0 | − | 19.6 |
| Phi-3-medium-128k-Inst | 42.9 | 57.0 | 0.0 | 24.0 | 3.0 | − | 25.4 |
| Qwen2.5-3B-Inst | 36.6 | 50.0 | 0.0 | 0.0 | 50.0 | − | 27.3 |
| Qwen2.5-7B-Inst | 45.2 | 60.0 | 0.0 | 2.0 | 32.0 | − | 27.8 |
| R1-Distill-Qwen2.5-7B$^{\mathcal{R}}$ | 7.2 | 12.0 | 3.0 | 4.0 | 55.0 | − | 16.3 |
| R1-Distill-Llama-3-8B$^{\mathcal{R}}$ | 30.0 | 31.0 | 82.0 | 32.0 | 83.0 | − | 51.6 |
| AI21-Jamba-1.5-Mini | 36.2 | 47.0 | 0.0 | 0.0 | 12.0 | − | 19.0 |
| Qwen2.5-32B-Inst | 74.5 | 81.0 | 29.0 | 65.0 | 96.0 | − | 68.4 |
| Qwen2.5-72B-Inst | 82.5 | 83.0 | 62.0 | 79.0 | 37.0 | − | 68.7 |
| Llama-3.1-70B-Inst | 65.6 | 75.0 | 45.0 | 90.0 | 89.0 | − | 72.9 |
| Llama-3.3-70B-Inst | 78.0 | 76.0 | 70.0 | 87.0 | 77.0 | − | 77.6 |
| R1-Distill-Qwen2.5-32B$^{\mathcal{R}}$ | 78.2 | 70.0 | 97.0 | 67.0 | 91.0 | − | 80.6 |
| R1-Distill-Llama-3-70B$^{\mathcal{R}}$ | 74.7 | 74.0 | 95.0 | 76.0 | 99.0 | − | 83.7 |
| Claude-3-5-sonnet-2410 | 82.1 | 65.0 | 77.0 | 100.0 | 68.0 | − | 78.4 |
| GPT-4o-mini-24-07 | 81.5 | 71.0 | 51.0 | 19.0 | 56.0 | − | 55.7 |
| GPT-4o-2024-08 | 87 | 90.0 | 98.0 | 100.0 | 99.0 | − | 94.8 |
| Gemini-1.5-flash-001 | 68.4 | 82.3 | 81.0 | 74.0 | 89.0 | − | 78.9 |
| Gemini-1.5-pro-001 | 81.3 | 81.6 | 97.0 | 92.0 | 94.0 | − | 89.2 |

Table 9: Performance of all tested LCLMs on the 0.5K set across tasks.

| Results on 2K set | | | | | | | |
| --- | --- | --- | --- | --- | --- | --- | --- |
| | HTML to TSV | Pseudocode to Code | Path Traversal | ToM Tracking | Countdown | Travel Planning | Average |
| Llama-3.2-1B-Inst | 0.5 | 0.0 | 0.0 | 0.0 | 0.0 | 0.0 | 0.1 |
| Llama-3.2-3B-Inst | 4.05 | 6.0 | 0.0 | 0.0 | 5.0 | 12.0 | 4.5 |
| Llama-3.1-8B-Inst | 28.95 | 28.0 | 0.0 | 0.0 | 12.0 | 55.0 | 20.7 |
| Llama-3-8B-ProLong-512k-Inst | 35.9 | 2.0 | 0.0 | 0.0 | 0.0 | 16.0 | 9.0 |
| Mistral-7B-Inst-v0.3 | 18.35 | 13.0 | 0.0 | 0.0 | 3.0 | 40.0 | 12.4 |
| Phi-3-small-128k-Inst | 13.2 | 17.0 | 0.0 | 0.0 | 9.0 | 30.0 | 11.5 |
| Phi-3-medium-128k-Inst | 29.3 | 13.0 | 0.0 | 0.0 | 1.0 | 30.0 | 12.2 |
| Qwen2.5-3B-Inst | 15.35 | 9.0 | 0.0 | 0.0 | 9.0 | 1.0 | 5.7 |
| Qwen2.5-7B-Inst | 32.1 | 31.0 | 0.0 | 0.0 | 36.0 | 39.0 | 23.0 |
| R1-Distill-Qwen2.5-7B$^{\mathcal{R}}$ | 0.9 | 0.0 | 0.0 | 0.0 | 43.0 | 2.0 | 7.7 |
| R1-Distill-Llama-3-8B$^{\mathcal{R}}$ | 19.4 | 5.0 | 7.0 | 3.0 | 69.0 | 44.0 | 24.6 |
| AI21-Jamba-1.5-Mini | 14.05 | 16.0 | 0.0 | 0.0 | 0.0 | 24.0 | 9.0 |
| Qwen2.5-32B-Inst | 46.9 | 65.0 | 0.0 | 8.0 | 87.0 | 95.0 | 50.3 |
| Qwen2.5-72B-Inst | 62.25 | 67.0 | 1.0 | 2.0 | 61.0 | 85.0 | 46.4 |
| Llama-3.1-70B-Inst | 58.75 | 59.0 | 0.0 | 45.0 | 92.0 | 93.0 | 58.0 |
| Llama-3.3-70B-Inst | 60.2 | 64.0 | 1.0 | 45.0 | 89.0 | 86.0 | 57.5 |
| R1-Distill-Qwen2.5-32B$^{\mathcal{R}}$ | 53.9 | 59.0 | 61.0 | 50.0 | 88.0 | 54.0 | 60.9 |
| R1-Distill-Llama-3-70B$^{\mathcal{R}}$ | 60.2 | 56.0 | 90.0 | 59.0 | 86.0 | 71.0 | 70.4 |
| Claude-3-5-sonnet-2410 | 66.9 | 73.0 | 35.0 | 1.0 | 78.0 | 91.0 | 57.5 |
| GPT-4o-mini-24-07 | 56.4 | 58.0 | 0.0 | 0.0 | 53.0 | 61.0 | 38.1 |
| GPT-4o-2024-08 | 76.4 | 84.0 | 77.0 | 77.0 | 95.0 | 91.0 | 83.4 |
| Gemini-1.5-flash-001 | 65.4 | 21.4 | 35.0 | 23.0 | 79.0 | 90.0 | 52.3 |
| Gemini-1.5-pro-001 | 75.25 | 50.0 | 96.0 | 71.0 | 84.0 | 100.0 | 79.4 |

Table 10: Performance of all tested LCLMs on the 2K set across tasks.

| | **Results on 8K set** | | | | | | |
|---|---|---|---|---|---|---|---|
| | HTML to TSV | Pseudocode to Code | Path Traversal | ToM Tracking | Countdown | Travel Planning | Average |
| Llama-3.2-1B-Inst | 0 | – | 0.0 | 0.0 | 0.0 | 0.0 | 0.0 |
| Llama-3.2-3B-Inst | 0.25 | – | 0.0 | 0.0 | 0.0 | 0.0 | 0.1 |
| Llama-3.1-8B-Inst | 23.4 | – | 0.0 | 0.0 | 3.0 | 0.0 | 5.3 |
| Llama-3-8B-ProLong-512k-Inst | 22.05 | – | 0.0 | 0.0 | 0.0 | 0.0 | 4.4 |
| Mistral-7B-Inst-v0.3 | 5.95 | – | 0.0 | 0.0 | 0.0 | 0.0 | 1.2 |
| Phi-3-small-128k-Inst | 4.25 | – | 0.0 | 0.0 | 1.0 | 0.0 | 1.1 |
| Phi-3-medium-128k-Inst | 12.25 | – | 0.0 | 0.0 | 0.0 | 0.0 | 2.5 |
| Qwen2.5-3B-Inst | 6.4 | – | 0.0 | 0.0 | 0.0 | 0.0 | 1.3 |
| Qwen2.5-7B-Inst | 16.95 | – | 0.0 | 0.0 | 2.0 | 0.0 | 3.8 |
| R1-Distill-Qwen2.5-7B$^{\mathcal{R}}$ | 0.0 | – | 0.0 | 0.0 | 12.0 | 0.0 | 2.4 |
| R1-Distill-Llama-3-8B$^{\mathcal{R}}$ | 8.4 | – | 0.0 | 0.0 | 26.0 | 3.0 | 7.5 |
| AI21-Jamba-1.5-Mini | 5.5 | – | 0.0 | 0.0 | 0.0 | 0.0 | 1.1 |
| Qwen2.5-32B-Inst | 25.4 | – | 0.0 | 0.0 | 55.0 | 5.0 | 17.1 |
| Qwen2.5-72B-Inst | 45.6 | – | 0.0 | 0.0 | 50.0 | 2.0 | 19.5 |
| Llama-3.1-70B-Inst | 47.05 | – | 0.0 | 0.0 | 59.0 | 15.0 | 24.2 |
| Llama-3.3-70B-Inst | 51.7 | – | 0.0 | 0.0 | 61.0 | 12.0 | 24.9 |
| R1-Distill-Qwen2.5-32B$^{\mathcal{R}}$ | 38.8 | – | 0.0 | 0.0 | 51.0 | 22.0 | 22.4 |
| R1-Distill-Llama-3-70B$^{\mathcal{R}}$ | 46.4 | – | 31.0 | 7.0 | 47.0 | 35.0 | 33.3 |
| Claude-3-5-sonnet-2410 | 52.1 | – | 1.0 | 0.0 | 34.0 | 23.0 | 22.0 |
| GPT-4o-mini-24-07 | 34.2 | – | 0.0 | 0.0 | 4.0 | 0.0 | 7.6 |
| GPT-4o-2024-08 | 65.45 | – | 34.0 | 0.0 | 67.0 | 24.0 | 38.1 |
| Gemini-1.5-flash-001 | 52.35 | – | 0.0 | 0.0 | 22.0 | 2.0 | 15.3 |
| Gemini-1.5-pro-001 | 70 | – | 81.0 | 28.0 | 46.0 | 45.0 | 54.0 |

Table 11: Performance of all tested LCLMs on the 8K set across tasks.

# F  Additional Data Examples

---

**Example F.1: example data point of HTML to TSV**

**HTML Page**
```
<html><head><title>UK Garage Finder |Search for Recommended Garages |UK Garage Search</title>
</head>
<body>
... ...
<div>
<div>1.</div>
<div>
<h3><a>AAK Autoservices</a></h3>
<h4><a>Units 62-63 John Wilson Business Park, Kent, CT5 3QT</a></h4>
</div>
</div>
<div>
<div>
<div>
<div><a></a></div>
<div>From 758 surveys 
100% recommendation score</div>
</div>
</div>
</div>
</div>
</div>
<div>
<div>
<div>
<div>
<div>
<p>Garage type: Independent Garage</p>
<p>Phone: 03301302600</p>
<p>Code: Service & Repair</p>
```

---

```
</div>
</div>
<div>
<div>
<div>
<div>
<p></p>
<div role="group">
<div role="group"><a>Info.</a></div>
<div role="group"><a>Rate </a></div>
</div>
</div>
</div>
</div>
</div>
<div>
<div>
<div>
<div>
... ...
</body>
... ...
</html>
```

**Target Information**
Based on the HTML webpage above about Garage search, extract the following properties from the items listed on the webpage: (1) Title; (2) Address; (3) Garage type; (4) Phone;
Only extract the rows where the garage type is "Independent Garage".

**TSV**
AAK Autoservices Units 62-63 John Wilson Business Park, Kent, CT5 3QT Independent Garage 03301302600
Autocare Centre Chepstow Unit 6 Bulwark Business Park Bulwark, Monmouthshire, NP16 5QZ Independent Garage 0330 1303262
Garej Arwyn Cyf Unit Ah Penygroes Ind Est, Gwynedd, LL54 6DB Independent Garage 03301300767
Horsham Motor Services Unit 3 Parsonage Farm Ind Est Parsonage Way, West Sussex, RH12 4AQ Independent Garage 03301301956
Robin Hood Garage Ltd Unit 1, Robin Hood Works Wakefield Road, Yorkshire, HD6 1PE Independent Garage 03301304478
Steve Wood Motors Ltd High Street, Lincolnshire, LN12 2HE Independent Garage 01507 441215
Chester Auto Services Bumpers Lane, Cheshire, CH1 4LT Independent Garage 01244 377747
Hardwick Motor Company Ltd Hardwick Lane, Nottinghamshire, NG17 5EJ Independent Garage 01623555512
Dukes AutoTech Greensplat, Cornwall, PL26 8XY Independent Garage 01726 337171
Nailsea Auto Electrical Unit 42, North Somerset, BS48 4NE Independent Garage 01275261005

## Example F.2: example data point of Pseudocode to Code

**Pseudocode Annotations**
define the main function
        str = string array of size 12
        n, i, j, ck = int
        read n then str[0]
        set str[1] to "vaporeon"
        set str[2] to "jolteon"
        set str[3] to "flareon"
        set str[4] to "espeon"
        set str[5] to "umbreon"
        set str[6] to "leafeon"
        set str[7] to "glaceon"

```
        set str[8] to "sylveon"
        for i = 1 to 8 inclusive
                if size of str[i] is n
                        set ck to 1
                        for j = 0 to n
                                if str[0][j] >= 'a' and str[0][j] <= 'z'
                                        if str[0][j] is not str[i][j]
                                                set ck to 0
                                                break
                                        end of the block
                                end of the block
                        end of the block
                        if ck is 1
                                print str[i]
                                return to indicate successful execution
                        end of the block
                end of the block
        end of the block
        print newline
        return to indicate successful execution
end of the block
```

**C++ Code**
```cpp
int main() {
        string str[12];
        int n, i, j, ck;
        cin >>n >>str[0];
        str[1] = "vaporeon";
        str[2] = "jolteon";
        str[3] = "flareon";
        str[4] = "espeon";
        str[5] = "umbreon";
        str[6] = "leafeon";
        str[7] = "glaceon";
        str[8] = "sylveon";
        for (i = 1; i <= 8; i++) {
                if (str[i].size() == n) {
                        ck = 1;
                        for (j = 0; j <n; j++) {
                                if (str[0][j] >= 'a' && str[0][j] <= 'z') {
                                        if (str[0][j] != str[i][j]) {
                                                ck = 0;
                                                break;
                                        }
                                }
                        }
                        if (ck == 1) {
                                cout <<str[i] <<endl;
                                return 0;
                        }
                }
        }
        cout <<endl;
        return 0;
}
```

## Example F.3: example data point of Path Traversal

**Problem Description**
In a hypothetical world, there are a number of cities. Each city has a one-way connection to only one other city via a specific transit method. The details of the cities are as follows:
Vladivostok is a lively city. You can travel from Vladivostok to Guildford by ferry.

Kendal is a lively city. You can travel from Kendal to Fort McMurray by plane.
Fort McMurray is a lively city. You can travel from Fort McMurray to Versailles by train.
Astrakhan is a lively city. You can travel from Astrakhan to Akron by bus.
Timisoara is a lively city. You can travel from Timisoara to Lafayette by train.
Bolzano is a lively city. You can travel from Bolzano to Terrace by bus.
Versailles is a lively city. You can travel from Versailles to West Valley City by ferry.
Lille is a lively city. You can travel from Lille to Abingdon by ferry.
Lafayette is a lively city. You can travel from Lafayette to Tucson by ferry.
Reno is a lively city. You can travel from Reno to Lafayette by plane.
... ...
Kamloops is a lively city. You can travel from Kamloops to Fort Collins by ferry.
Livorno is a lively city. You can travel from Livorno to Colorado Springs by train.
Vladikavkaz is a lively city. You can travel from Vladikavkaz to Bromsgrove by plane.
Medicine Hat is a lively city. You can travel from Medicine Hat to Tallinn by ferry.
Sandhurst is a lively city. You can travel from Sandhurst to Erfurt by ferry.
Tucson is a lively city. You can travel from Tucson to Gateshead by train.
Chilliwack is a lively city. You can travel from Chilliwack to Naperville by bus.
Sacramento is a lively city. You can travel from Sacramento to Kendal by ferry.
Folkestone is a lively city. You can travel from Folkestone to Reno by ferry.
Bournemouth is a lively city. You can travel from Bournemouth to Cornwall by plane.

Now find the route from Lille to Bromsgrove based on the information above.

**Target Route**
From Lille, take a ferry to Abingdon.
From Abingdon, take a bus to Augsburg.
From Augsburg, take a plane to Lecce.
From Lecce, take a plane to Vladivostok.
From Vladivostok, take a ferry to Guildford.
From Guildford, take a ferry to Gelsenkirchen.
From Gelsenkirchen, take a train to Kamloops.
From Kamloops, take a ferry to Fort Collins.
From Fort Collins, take a train to Basingstoke.
From Basingstoke, take a plane to Medicine Hat.
... ...
From Sacramento, take a ferry to Kendal.
From Kendal, take a plane to Fort McMurray.
From Fort McMurray, take a train to Versailles.
From Versailles, take a ferry to West Valley City.
From West Valley City, take a plane to Crewe.
From Crewe, take a plane to Worthing.
From Worthing, take a ferry to Bournemouth.
From Bournemouth, take a plane to Cornwall.
From Cornwall, take a ferry to Vladikavkaz.
From Vladikavkaz, take a plane to Bromsgrove.

### Example F.4: example data point of Theory-of-Mind Tracking

**Story**
You'll see a story about object placement. Each story involves four components: Agents, Objects, Rooms, and Containers. Given a question about an (agent, object) pair, your task is to track the locations and beliefs in stories about object placement asked in the question.
Step 0: Leon is in the playroom; Carol is in the pantry; the band-aid is on the playroom's pedestal; the tweezers is on the playroom's pedestal.
Step 1: Leon moves to the pantry.
Step 2: Carol moves to the playroom.
Step 3: Leon moves to the playroom.
Step 4: Carol moves to the pantry.
Step 5: Leon moves the band-aid to the playroom's rack.
Step 6: Carol moves to the playroom.
Step 7: Leon moves to the pantry, and moves the tweezers to the pantry's pedestal.

Step 8: Carol moves to the pantry, and moves the band-aid to the pantry's pedestal.
Step 9: Leon moves the tweezers to the pantry's rack.
... ...
Step 30: Carol enters the playroom.
Step 31: Leon moves to the playroom, and moves the band-aid to the playroom's rack.
Step 32: Carol moves to the pantry, and moves the tweezers to the pantry's pedestal.
Step 33: Leon moves to the pantry.
Step 34: Carol moves to the playroom, and moves the tweezers to the playroom's pedestal.
Step 35: Leon moves to the playroom.
Step 36: Carol leaves the room he was in.
Step 37: Leon moves to the pantry, and moves the tweezers to the pantry's rack.
Step 38: Carol enters the playroom.
Step 39: Leon leaves the room he was in.
Step 40: Carol moves to the pantry, and moves the band-aid to the pantry's rack.

Where does Carol believe the tweezers is?

---

**Target Output**
Carol's belief on the tweezers:
- Step 0: (initial state) {Carol location: pantry; tweezers location: playroom's pedestal; Carol sees tweezers: False; Carol's belief on tweezers: None}
- Step 1: (Carol location unchanged; tweezers location unchanged) {Carol location: pantry; tweezers location: playroom's pedestal; Carol sees tweezers: False; Carol's belief on tweezers: None}
- Step 2: (Carol location changed; tweezers location unchanged) {Carol location: playroom; tweezers location: playroom's pedestal; Carol sees tweezers: True; Carol's belief on tweezers: playroom's pedestal}
- Step 3: (Carol location unchanged; tweezers location unchanged) {Carol location: playroom; tweezers location: playroom's pedestal; Carol sees tweezers: True; Carol's belief on tweezers: playroom's pedestal}
- Step 4: (Carol location changed; tweezers location unchanged) {Carol location: pantry; tweezers location: playroom's pedestal; Carol sees tweezers: False; Carol's belief on tweezers: playroom's pedestal}
- Step 5: (Carol location unchanged; tweezers location unchanged) {Carol location: pantry; tweezers location: playroom's pedestal; Carol sees tweezers: False; Carol's belief on tweezers: playroom's pedestal}
- Step 6: (Carol location changed; tweezers location unchanged) {Carol location: playroom; tweezers location: playroom's pedestal; Carol sees tweezers: True; Carol's belief on tweezers: playroom's pedestal}
- Step 7: (Carol location unchanged; tweezers location changed) {Carol location: playroom; tweezers location: pantry's pedestal; Carol sees tweezers: False; Carol's belief on tweezers: playroom's pedestal}
- Step 8: (Carol location changed; tweezers location unchanged) {Carol location: pantry; tweezers location: pantry's pedestal; Carol sees tweezers: True; Carol's belief on tweezers: pantry's pedestal}
- Step 9: (Carol location unchanged; tweezers location changed) {Carol location: pantry; tweezers location: pantry's rack; Carol sees tweezers: True; Carol's belief on tweezers: pantry's rack}
... ...
- Step 30: (Carol location changed; tweezers location unchanged) {Carol location: playroom; tweezers location: playroom's pedestal; Carol sees tweezers: True; Carol's belief on tweezers: playroom's pedestal}
- Step 31: (Carol location unchanged; tweezers location unchanged) {Carol location: playroom; tweezers location: playroom's pedestal; Carol sees tweezers: True; Carol's belief on tweezers: playroom's pedestal}
- Step 32: (Carol location changed; tweezers location changed) {Carol location: pantry; tweezers location: pantry's pedestal; Carol sees tweezers: True; Carol's belief on tweezers: pantry's pedestal}
- Step 33: (Carol location unchanged; tweezers location unchanged) {Carol location: pantry; tweezers location: pantry's pedestal; Carol sees tweezers: True; Carol's belief on tweezers: pantry's pedestal}
- Step 34: (Carol location changed; tweezers location changed) {Carol location: playroom; tweezers location: playroom's pedestal; Carol sees tweezers: True; Carol's belief on tweezers: playroom's pedestal}
- Step 35: (Carol location unchanged; tweezers location unchanged) {Carol location: playroom; tweezers location: playroom's pedestal; Carol sees tweezers: True; Carol's belief on tweezers:

playroom's pedestal}
- Step 36: (Carol location changed; tweezers location unchanged) {Carol location: None; tweezers location: playroom's pedestal; Carol sees tweezers: False; Carol's belief on tweezers: playroom's pedestal}
- Step 37: (Carol location unchanged; tweezers location changed) {Carol location: None; tweezers location: pantry's rack; Carol sees tweezers: False; Carol's belief on tweezers: playroom's pedestal}
- Step 38: (Carol location changed; tweezers location unchanged) {Carol location: playroom; tweezers location: pantry's rack; Carol sees tweezers: False; Carol's belief on tweezers: playroom's pedestal}
- Step 39: (Carol location unchanged; tweezers location unchanged) {Carol location: playroom; tweezers location: pantry's rack; Carol sees tweezers: False; Carol's belief on tweezers: playroom's pedestal}
- Step 40: (Carol location changed; tweezers location unchanged) {Carol location: pantry; tweezers location: pantry's rack; Carol sees tweezers: True; Carol's belief on tweezers: pantry's rack}
Final Answer: the pantry's rack

## Example F.5: example data point of Countdown

**Problem**
You will be given four numbers and a target number, your task is to find a way to use all four numbers exactly once, along with the basic operations (+, -, *, /), to reach the target number.
Numbers: [44, 48, 35, 6]
Target: 25

**Solving Procedure**
Initial number set: [44, 48, 35, 6], target: 25. Options for choosing two numbers: [(44, 48), (44, 35), (44, 6), (48, 35), (48, 6), (35, 6)].
 |- Pick two numbers (44, 48) (numbers left: [35, 6]). Try possible operations.
   |- Try 48 + 44 = 92. Add 92 to the number set. Current number set: [92, 35, 6], target: 25. Options for choosing two numbers: [(92, 35), (92, 6), (35, 6)].
     |- Pick two numbers (92, 35) (numbers left: [6]). Try possible operations.
       |- Try 92 + 35 = 127. Add 127 to the number set. Current number set: [127, 6], target: 25, just two numbers left.
         |- Try 127 + 6 = 133. Evaluate 133 != 25, drop this branch.
         |- Try 127 - 6 = 121. Evaluate 121 != 25, drop this branch.
         |- Try 127 * 6 = 762. Evaluate 762 != 25, drop this branch.
         |- Try 127 / 6 = 21.2. 21.2 is a decimal, drop this branch.
       |- Try 92 - 35 = 57. Add 57 to the number set. Current number set: [57, 6], target: 25, just two numbers left.
         |- Try 57 + 6 = 63. Evaluate 63 != 25, drop this branch.
         |- Try 57 - 6 = 51. Evaluate 51 != 25, drop this branch.
         |- Try 57 * 6 = 342. Evaluate 342 != 25, drop this branch.
         |- Try 57 / 6 = 9.5. 9.5 is a decimal, drop this branch.
       |- Try 92 * 35 = 3220. 3220 exceeds the maximum intermediate result, drop this branch.
       |- Try 92 / 35 = 2.6. 2.6 is a decimal, drop this branch.
     |- Pick two numbers (92, 6) (numbers left: [35]). Try possible operations.
       |- Try 92 + 6 = 98. Add 98 to the number set. Current number set: [98, 35], target: 25, just two numbers left.
         |- Try 98 + 35 = 133. Evaluate 133 != 25, drop this branch.
         |- Try 98 - 35 = 63. Evaluate 63 != 25, drop this branch.
         |- Try 98 * 35 = 3430. 3430 exceeds the maximum intermediate result, drop this branch.
         |- Try 98 / 35 = 2.8. 2.8 is a decimal, drop this branch.
       |- Try 92 - 6 = 86. Add 86 to the number set. Current number set: [86, 35], target: 25, just two numbers left.
         |- Try 86 + 35 = 121. Evaluate 121 != 25, drop this branch.
         |- Try 86 - 35 = 51. Evaluate 51 != 25, drop this branch.
         |- Try 86 * 35 = 3010. 3010 exceeds the maximum intermediate result, drop this branch.
         |- Try 86 / 35 = 2.5. 2.5 is a decimal, drop this branch.
       |- Try 92 * 6 = 552. Add 552 to the number set. Current number set: [552, 35], target: 25, just two numbers left.
         |- Try 552 + 35 = 587. Evaluate 587 != 25, drop this branch.

|- Try 552 - 35 = 517. Evaluate 517 != 25, drop this branch.
    |- Try 552 * 35 = 19320. 19320 exceeds the maximum intermediate result, drop this branch.
    |- Try 552 / 35 = 15.8. 15.8 is a decimal, drop this branch.
  |- Try 92 / 6 = 15.3. 15.3 is a decimal, drop this branch.
 |- Pick two numbers (35, 6) (numbers left: [92]). Try possible operations.
  |- Try 35 + 6 = 41. Add 41 to the number set. Current number set: [41, 92], target: 25, just two numbers left.
    |- Try 92 + 41 = 133. Evaluate 133 != 25, drop this branch.
    |- Try 92 - 41 = 51. Evaluate 51 != 25, drop this branch.
    |- Try 92 * 41 = 3772. 3772 exceeds the maximum intermediate result, drop this branch.
    |- Try 92 / 41 = 2.2. 2.2 is a decimal, drop this branch.
  |- Try 35 - 6 = 29. Add 29 to the number set. Current number set: [29, 92], target: 25, just two numbers left.
    |- Try 92 + 29 = 121. Evaluate 121 != 25, drop this branch.
    |- Try 92 - 29 = 63. Evaluate 63 != 25, drop this branch.
    |- Try 92 * 29 = 2668. 2668 exceeds the maximum intermediate result, drop this branch.
    |- Try 92 / 29 = 3.2. 3.2 is a decimal, drop this branch.
  |- Try 35 * 6 = 210. Add 210 to the number set. Current number set: [210, 92], target: 25, just two numbers left.
    |- Try 210 + 92 = 302. Evaluate 302 != 25, drop this branch.
    |- Try 210 - 92 = 118. Evaluate 118 != 25, drop this branch.
    |- Try 210 * 92 = 19320. 19320 exceeds the maximum intermediate result, drop this branch.
    |- Try 210 / 92 = 2.3. 2.3 is a decimal, drop this branch.
  |- Try 35 / 6 = 5.8. 5.8 is a decimal, drop this branch.
|- Try 48 - 44 = 4. Add 4 to the number set. Current number set: [4, 35, 6], target: 25. Options for choosing two numbers: [(4, 35), (4, 6), (35, 6)].
 |- Pick two numbers (4, 35) (numbers left: [6]). Try possible operations.
  |- Try 35 + 4 = 39. Add 39 to the number set. Current number set: [39, 6], target: 25, just two numbers left.
    |- Try 39 + 6 = 45. Evaluate 45 != 25, drop this branch.
    |- Try 39 - 6 = 33. Evaluate 33 != 25, drop this branch.
    |- Try 39 * 6 = 234. Evaluate 234 != 25, drop this branch.
    |- Try 39 / 6 = 6.5. 6.5 is a decimal, drop this branch.
  |- Try 35 - 4 = 31. Add 31 to the number set. Current number set: [31, 6], target: 25, just two numbers left.
    |- Try 31 + 6 = 37. Evaluate 37 != 25, drop this branch.
    |- Try 31 - 6 = 25. Evaluate 25 == 25, target found!

Now we have found the target, let's trace back the solution:
Final step: 31 - 6 = 25
The step before: 35 - 4 = 31
The first step: 48 - 44 = 4

Output the solution in the required format:
<Solution>
48 - 44 = 4
35 - 4 = 31
31 - 6 = 25
</Solution>

## Example F.6: example data point of Travel Planning

**Problem**
You plan to visit 5 European cities for 20 days in total. You only take direct flights to commute between cities. You want to spend 7 days in Hamburg. You would like to visit Munich for 6 days. You want to spend 2 days in Manchester. You plan to visit relatives in Manchester between day 19 and day 20. You plan to stay in Lyon for 2 days. From day 13 to day 14, there is a annual show you want to attend in Lyon. You would like to visit Split for 7 days.

Here are the cities that have direct flights:
from Split to Munich, from Munich to Manchester, from Hamburg to Manchester, from Hamburg to Munich, from Split to Lyon, from Lyon to Munich, from Hamburg to Split, from Manchester to

Split.

Find a trip plan of visiting the cities for 20 days by taking direct flights to commute between them.

**Solving Procedure**
Read the requirements and identify the cities that have fixed schedules and the cities that need to be arranged.
* City: Hamburg, Duration: 7 days.
* City: Munich, Duration: 6 days.
* City: Manchester, Duration: 2 days, Fixed Schedule: Day 19 - 20.
* City: Lyon, Duration: 2 days, Fixed Schedule: Day 13 - 14.
* City: Split, Duration: 7 days.

Cities that have fixed schedules (sorted by their arrival days):
* City: Lyon, Fixed Schedule: Day 13 - 14.
* City: Manchester, Fixed Schedule: Day 19 - 20.

Cities needing arrangement:
* City: Hamburg, Duration: 7 days.
* City: Munich, Duration: 6 days.
* City: Split, Duration: 7 days.

Current day: 1. Current plan: [].
Check whether the city with an arrival day of Day 1 - is fixed.
No. Consider possible options from cities needing arrangement: [Hamburg, Munich, Split] and explore these options in order.
 |- Try arranging to visit Hamburg from Day 1. Duration: 7 days. Schedule: Day 1 - 7.
 |- Check for direct flight from the starting point to Hamburg.
 |- Yes.
 |- Check whether this arrangement is compatible with the next fixed schedule after Day 1: Lyon (Day 13 - 14).
 |- The departure day of Hamburg is Day 7. The arrival day of Lyon is Day 13. Day 7 is not later than (<=) Day 13. This arrangement is compatible.
 |- This arrangement is feasible for now. Continue to arrange the rest of the plan.
    |- Current day: 7. Current plan: [Hamburg].
    |- Check whether the city with an arrival day of Day 7 - is fixed.
    |- No. Consider possible options from cities needing arrangement: [Munich, Split] and explore these options in order.
       |- Try arranging to visit Munich from Day 7. Duration: 6 days. Schedule: Day 7 - 12.
       |- Check for direct flight from Hamburg to Munich.
       |- Yes.
       |- Check whether this arrangement is compatible with the next fixed schedule after Day 7: Lyon (Day 13 - 14).
       |- The departure day of Munich is Day 12. The arrival day of Lyon is Day 13. Day 12 is not later than (<=) Day 13. This arrangement is compatible.
       |- This arrangement is feasible for now. Continue to arrange the rest of the plan.
          |- Current day: 12. Current plan: [Hamburg, Munich].
          |- Check whether the city with an arrival day of Day 12 - is fixed.
          |- No. Consider possible options from cities needing arrangement: [Split] and explore these options in order.
             |- Try arranging to visit Split from Day 12. Duration: 7 days. Schedule: Day 12 - 18.
             |- Check for direct flight from Munich to Split.
             |- No. Drop this branch.
             |- Fail to arrange any option on day 12 in the current arrangement. Drop this branch.
       |- Try arranging to visit Split from Day 7. Duration: 7 days. Schedule: Day 7 - 13.
       |- Check for direct flight from Hamburg to Split.
       |- Yes.
       |- Check whether this arrangement is compatible with the next fixed schedule after Day 7: Lyon (Day 13 - 14).
       |- The departure day of Split is Day 13. The arrival day of Lyon is Day 13. Day 13 is not later than (<=) Day 13. This arrangement is compatible.
       |- This arrangement is feasible for now. Continue to arrange the rest of the plan.

|- Current day: 13. Current plan: [Hamburg, Split].
|- Check whether the city with an arrival day of Day 13 - is fixed.
|- Yes. The city with an arrival day of Day 13 - is fixed: Lyon.
  |- Try arranging to visit Lyon from Day 13. Duration: 2 days. Schedule: Day 13 - 14.
  |- Check for direct flight from Split to Lyon.
  |- Yes.
  |- Check whether this arrangement is compatible with the next fixed schedule after Day 13:
Manchester (Day 19 - 20).
  |- The departure day of Lyon is Day 14. The arrival day of Manchester is Day 19. Day 14 is
not later than (<=) Day 19. This arrangement is compatible.
  |- This arrangement is feasible for now. Continue to arrange the rest of the plan.
    |- Current day: 14. Current plan: [Hamburg, Split, Lyon].
    |- Check whether the city with an arrival day of Day 14 - is fixed.
    |- No. Consider possible options from cities needing arrangement: [Munich] and explore
these options in order.
      |- Try arranging to visit Munich from Day 14. Duration: 6 days. Schedule: Day 14 - 19.
      |- Check for direct flight from Lyon to Munich.
      |- Yes.
      |- Check whether this arrangement is compatible with the next fixed schedule after Day
14: Manchester (Day 19 - 20).
      |- The departure day of Munich is Day 19. The arrival day of Manchester is Day 19. Day
19 is not later than (<=) Day 19. This arrangement is compatible.
      |- This arrangement is feasible for now. Continue to arrange the rest of the plan.
        |- Current day: 19. Current plan: [Hamburg, Split, Lyon, Munich].
        |- Check whether the city with an arrival day of Day 19 - is fixed.
        |- Yes. The city with an arrival day of Day 19 - is fixed: Manchester.
         |- Try arranging to visit Manchester from Day 19. Duration: 2 days. Schedule: Day 19 -
20.
        |- Check for direct flight from Munich to Manchester.
        |- Yes.
        |- Check whether this arrangement is compatible with the next fixed schedule after
Day 19: None.
        |- No following fixed schedules. This arrangement is compatible.
        |- This arrangement is feasible for now. Continue to arrange the rest of the plan.
         |- Current day: 20. Current plan: [Hamburg, Split, Lyon, Munich, Manchester].
         |- All 5 cities are arranged. Complete plan is found!

Output the plan in the required format:
<Plan>
**Day 1-7:** Arriving in Hamburg and visit Hamburg for 7 days.
**Day 7:** Fly from Hamburg to Split.
**Day 7-13:** Visit Split for 7 days.
**Day 13:** Fly from Split to Lyon.
**Day 13-14:** Visit Lyon for 2 days.
**Day 14:** Fly from Lyon to Munich.
**Day 14-19:** Visit Munich for 6 days.
**Day 19:** Fly from Munich to Manchester.
**Day 19-20:** Visit Manchester for 2 days.
</Plan>

# G   Example Error Cases

## G.1   HTML to TSV

Example G.1 shows an example of GPT-4o failing to copy all rows from the HTML input of
an 8K test set without filters. Note that the model is only able to extract the first 5 rows and
then skip the remaining rows.

**Example G.1: An example of copying error without filters for HTML to TSV**

**HTML Page**
<html><head>
<title>January 2021 - Page 2 of 2 - Juicers Zones</title>
</head>
<body>
<div>
<a>Skip to content</a>
<header>
<div>
<div>
<div>
<div>
<div>
<h3>
<a title="Juicers Zones">Juicers Zones</a>
</h3>
</div></div><div>
<nav role="navigation">
<p>
Menu
</p>
<div><ul><li><a>Juice Zone</a></li>
<li><a>Privacy Policy</a></li>
<li><a>Contact</a></li>
</ul></div></nav>
</div></div></div></div><div></div>
<div>
<div>
<div>
<h1>Month: January 2021</h1>
</div>
</div>
</div>
</header>
<div>
<div>
<div>
<div>
<article>
<header>
<h2>
<a title="Lidar Pasadena..">Lidar Pasadena..</a>
</h2></header>
<div>
<p>Local Surveyor Pasadena Hiring a land surveyor is something many people do only once or twice in their lives, so they don't have plenty of experience when determining who to hire. Hiring a surveyor is, in many ways, like hiring</p>
</div>
<footer><div>
<a>Sheree</a>
<a title="5:01 pm"><time>January 14, 2021</time></a><a>Juice Zone</a>

<a>Read more</a>... ...
</body>
... ...
</html>

**Target Information**
Based on the HTML webpage above about Articles, extract the following properties from the items listed on the webpage: (1) Title; (2) Author; (3) Date; (4) Description;

**Ground Truth TSV**
Lidar Pasadena.. Sheree January 14, 2021 Local Surveyor Pasadena Hiring a land surveyor is something many people do only once or twice in their lives, so they don't have plenty of experience when determining who to hire. Hiring a surveyor is, in many ways, like hiring
Happy Valentines Day Wishes – Remarkable Benefits.. Sheree January 14, 2021 Valentine's Day is recognized every year by people and allows men and women to show the love they have for their beloved. Even though all have special days, this particular day is accepted as being a special day to espouse
WWE Corporate Office Headquarters – Why Is This Significant.. Sheree January 12, 2021 The WWE Divas are some of the hottest ladies on the planet today, as well as many of the most athletic entertainers. The latest divas include the likes of Candice Michelle, Torrie Wilson, Melina Perez, Maria Kanellis, Victoria, Jillian Hall
9 Skincare Tips – Check This Out.. Sheree January 11, 2021 Beauty Tips Blog Taking care of your body means getting regular exercise, and good nourishment nevertheless in the rushed fast paced lifestyles in our 21st century, and many holding down demanding careers, this may not be quite simple to do.
Paraiba World Ltd – Just Published Sheree January 9, 2021 Automated buying and selling platforms driven by configurable crypto buying and selling crawlers are increasing. Exactly what are the pros and cons of those platforms, that are the marketplace leaders – and must you make use of one? The maturing
Kroger Headquarters Contacts – Useful Information.. Sheree January 8, 2021 You can either get your Kroger weekly advertisements every weekend break and let it sit there unused or you can put it to great use and really start saving lots of money. Me, I prefer to make the most of
... ...
[43 more rows]
... ...
**Model Output TSV**
Lidar Pasadena.. Sheree January 14, 2021 Local Surveyor Pasadena Hiring a land surveyor is something many people do only once or twice in their lives, so they don't have plenty of experience when determining who to hire. Hiring a surveyor is, in many ways, like hiring
Happy Valentines Day Wishes – Remarkable Benefits.. Sheree January 14, 2021 Valentine's Day is recognized every year by people and allows men and women to show the love they have for their beloved. Even though all have special days, this particular day is accepted as being a special day to espouse
WWE Corporate Office Headquarters – Why Is This Significant.. Sheree January 12, 2021 The WWE Divas are some of the hottest ladies on the planet today, as well as many of the most athletic entertainers. The latest divas include the likes of Candice Michelle, Torrie Wilson, Melina Perez, Maria Kanellis, Victoria, Jillian Hall
9 Sk

Example G.2 shows an example of GPT-4o extracting wrong rows that do not satisfy the given filtering instruction from a 2K test example with filters. While the model is prompted to extract rows where the type of the household is strictly equal to "2 Beds" (partial match such as "1-2 Beds" is not valid), the model output still includes rows where the type of the household is "Studio" or "1-3 Beds".

---

**Example G.2: An example of filtering error with filters for HTML to TSV**

**HTML Page**
<html><head><title>Apartments for Rent in 90210, Beverly Hills, CA |Apartment Finder</title>
</head>
<body>
<div>
<header>
<div/>
<div>
<div>
<a>
<div/>
<div/>
<div/>
</a>

```
</div>
<div>
<nav>
<h2>
Header Navigation Links
</h2>
<ul>
<li>
<form>
<div>
<div role="combobox">
<label>Search label</label>
<input placeholder="City, Zip, Neighborhood, Address, Property Name" type="text"/>
<div/>
</div>
</div>
<div role="listbox"/>
<div/>
</form>
</li>
<li>
<a>

</a>
</li>
<li>
... ...
<div>
<h2>
<a title="609 N Doheny Dr">
609 N Doheny Dr
</a>
</h2>
<address title="609 N Doheny Dr, Beverly Hills, CA 90210">
609 N Doheny Dr, Beverly Hills, CA 90210
</address>
<div>
<div>

$4,500


|

2 Beds


</div>
</div>
</div>
<div>
<button title="Send Message">
Send Message
Email Property
</button>
|
<a>
Call Now
(657) 289-6985
</a>
</div>
</div>
</article><article>
<div>
```

```
<div>
<button/>

1
/
48

<div>
<button role="button" type="button"/>
<button role="button" type="button"/>
... ...
</body>
... ...
</html>
```

**Target Information**
Based on the HTML webpage above about Real estate search, extract the following properties from the items listed on the webpage: (1) Name; (2) Detailed address, including zip code; (3) Rent; (4) Type of the household (number of beds/studio); (5) Phone;
Only extract the rows where the type of the household is strictly equal to **"2 Beds"**.

---

**Ground Truth TSV**
609 N Doheny Dr 609 N Doheny Dr, Beverly Hills, CA 90210 $4,500 2 Beds (657) 289-6985 2 Wks. Ago
237 S Doheny Dr 237 S Doheny Dr, Beverly Hills, CA 90211 $3,500 2 Beds (805) 833-6612 8 Hrs. Ago
135 N Doheny Dr 135 N Doheny Dr, West Hollywood, CA 90048 $3,600 - $3,995 2 Beds (858) 704-2956 10 Hrs. Ago
8871 Burton Way 8871 Burton Way, West Hollywood, CA 90048 $3,500 2 Beds (424) 566-8685 1 Day Ago
140 S Crescent Dr 140 S Crescent Dr, Beverly Hills, CA 90212 $5,500 2 Beds (424) 380-6264 2 Days Ago
137 N Wetherly Dr 137 N Wetherly Dr, Los Angeles, CA 90048 $3,100 - $3,200 2 Beds (562) 553-7450 2 Days Ago
9005 Cynthia St 9005 Cynthia St, West Hollywood, CA 90069 $3,600 2 Beds (805) 833-6586 3 Days Ago
310 S Almont Dr 310 S Almont Dr, Los Angeles, CA 90048 $2,950 2 Beds (626) 701-8056 3 Days Ago
148 S Maple Dr 148 S Maple Dr, Beverly Hills, CA 90212 $2,995 2 Beds (747) 788-4237 4 Days Ago
9061 Keith Ave 9061 Keith Ave, West Hollywood, CA 90069 $3,100 2 Beds (424) 380-5750 4 Days Ago
200 S Spalding Dr 200 S Spalding Dr, Beverly Hills, CA 90212 $3,200 2 Beds (844) 653-6049 4 Days Ago
462 N Almont Dr 462 N Almont Dr, West Hollywood, CA 90048 $3,600 2 Beds (657) 234-6681 6 Days Ago
131 N Wetherly Dr 131 N Wetherly Dr, West Hollywood, CA 90048 $5,150 2 Beds (657) 224-3710 1 Wk. Ago
9011 Rangely Ave 9011 Rangely Ave, West Hollywood, CA 90048 $5,000 2 Beds (661) 388-5353 1 Wk. Ago

**Model Output TSV**
447 N Oakhurst Dr 447 N Oakhurst Dr, Beverly Hills, CA 90210 $1,650 Studio (424) 313-1711 6 Days Ago
328 N Maple Dr 328 N Maple Dr, Beverly Hills, CA 90210 $4,300 - $7,000 1 - 2 Beds (760) 536-5804 6 Days Ago
609 N Doheny Dr 609 N Doheny Dr, Beverly Hills, CA 90210 $4,500 2 Beds (657) 289-6985 2 Wks. Ago
237 S Doheny Dr 237 S Doheny Dr, Beverly Hills, CA 90211 $3,500 2 Beds (805) 833-6612 8 Hrs. Ago
135 N Doheny Dr 135 N Doheny Dr, West Hollywood, CA 90048 $3,600 - $3,995 2 Beds (858) 704-2956 10 Hrs. Ago
220 S Doheny Dr 220 S Doheny Dr, Beverly Hills, CA 90211 $2,295 - $2,695 1 - 2 Beds (763) 284-5475 1 Day Ago
115 N Doheny Dr 115 N Doheny Dr, West Hollywood, CA 90048 $1,700 - $3,000 Studio - 2 Beds (562) 349-1380 1 Day Ago
153 S Palm Dr 153 S Palm Dr, Beverly Hills, CA 90212 $5,495 - $7,495 2 - 3 Beds (424) 424-0578 1

Day Ago
137 N Wetherly Dr 137 N Wetherly Dr, Los Angeles, CA 90048 $3,100 - $3,200 2 Beds (562) 553-7450 2 Days Ago
9480 Charleville Blvd 9480 Charleville Blvd, Beverly Hills, CA 90212 $3,500 - $5,000 1 - 3 Beds (844) 900-0934 2 Days Ago
9005 Cynthia St 9005 Cynthia St, West Hollywood, CA 90069 $3,600 2 Beds (805) 833-6586 3 Days Ago
310 S Almont Dr 310 S Almont Dr, Los Angeles, CA 90048 $2,950 2 Beds (626) 701-8056 3 Days Ago
9061 Keith Ave 9061 Keith Ave, West Hollywood, CA 90069 $3,100 2 Beds (424) 380-5750 4 Days Ago
200 S Spalding Dr 200 S Spalding Dr, Beverly Hills, CA 90212 $3,200 2 Beds (844) 653-6049 4 Days Ago
9011 Rangely Ave 9011 Rangely Ave, West Hollywood, CA 90048 $5,000 2 Beds (661) 388-5353 1 Wk. Ago

## G.2 Theory-of-Mind Tracking

Example G.3 shows an example of GPT-4o incorrectly tracking the location of objects after long range for a 2K example of the theory-of-mind tracking task. The model incorrectly identifies the location of bobby pin at step 28 and carries the error on to later steps.

**Example G.3: An example of incorrectly updating search states for Countdown**

**Story**
You'll see a story about object placement. Each story involves four components: Agents, Objects, Rooms, and Containers. Given a question about an (agent, object) pair, your task is to track the locations and beliefs in stories about object placement asked in the question.
Step 0: Kevin is in the living room; Amanda is in the craft room; the USB cable is on the living room's tv stand; the rubber band is on the living room's stool.
Step 1: Kevin moves to the craft room.
Step 2: Amanda moves to the living room.
Step 3: Amanda moves to the craft room, and moves the rubber band to the craft room's tv stand.
Step 4: Amanda moves the rubber band to the craft room's stool.
Step 5: Kevin moves the rubber band to the craft room's tv stand.
Step 6: Amanda moves to the living room, and moves the rubber band to the living room's tv stand.
Step 7: Kevin moves to the living room.
Step 8: Amanda moves the USB cable to the living room's stool.
Step 9: Kevin moves to the craft room.
Step 10: Amanda leaves the room he was in.
Step 11: Kevin moves to the living room.
Step 12: Amanda enters the craft room.
Step 13: Kevin leaves the room he was in.
Step 14: Amanda moves to the living room.
Step 15: Kevin enters the living room.
Step 16: Amanda moves to the craft room, and moves the USB cable to the craft room's stool.
Step 17: Kevin moves to the craft room, and moves the rubber band to the craft room's tv stand.
Step 18: Amanda moves to the living room, and moves the USB cable to the living room's tv stand.
Step 19: Kevin moves to the living room, and moves the rubber band to the living room's tv stand.
Step 20: Amanda moves to the craft room, and moves the USB cable to the craft room's stool.
Step 21: Kevin moves to the craft room.
Step 22: Amanda moves to the living room, and moves the USB cable to the living room's stool.
Step 23: Kevin moves to the living room.
Step 24: Amanda moves the USB cable to the living room's tv stand.
Step 25: Kevin moves the USB cable to the living room's stool.
Step 26: Amanda moves the rubber band to the living room's stool.
Step 27: Kevin moves the USB cable to the living room's tv stand.
Step 28: Amanda leaves the room he was in.
Step 29: Kevin leaves the room he was in.
Step 30: Amanda enters the living room.
Step 31: Kevin enters the craft room.
Step 32: Amanda leaves the room he was in.

Step 33: Kevin moves to the living room.
Step 34: Amanda enters the living room.
Step 35: Kevin moves the rubber band to the living room's tv stand.
Step 36: Amanda moves to the craft room.
Step 37: Kevin moves to the craft room, and moves the USB cable to the craft room's tv stand.
Step 38: Amanda moves to the living room.
Step 39: Kevin moves to the living room, and moves the USB cable to the living room's stool.
Step 40: Amanda leaves the room he was in.
Where does Amanda believe the USB cable is?

**Ground Truth**
Gwendolyn's belief on the bobby pin:
- Step 0: (initial state) {Gwendolyn location: closet; bobby pin location: pantry's ledge; Gwendolyn sees bobby pin: False; Gwendolyn's belief on bobby pin: None}
... ...
- Step 27: (Gwendolyn location unchanged; bobby pin location unchanged) {Gwendolyn location: closet; bobby pin location: closet's ledge; Gwendolyn sees bobby pin: True; Gwendolyn's belief on bobby pin: closet's ledge}
- Step 28: (Gwendolyn location unchanged; bobby pin location unchanged) {Gwendolyn location: closet; bobby pin location: closet's ledge; Gwendolyn sees bobby pin: True; Gwendolyn's belief on bobby pin: closet's ledge}
- Step 29: (Gwendolyn location changed; bobby pin location unchanged) {Gwendolyn location: None; bobby pin location: closet's ledge; Gwendolyn sees bobby pin: False; Gwendolyn's belief on bobby pin: closet's ledge}
... ...
**Model Output**
Gwendolyn's belief on the bobby pin:
- Step 0: (initial state) {Gwendolyn location: closet; bobby pin location: pantry's ledge; Gwendolyn sees bobby pin: False; Gwendolyn's belief on bobby pin: None}
... ...
- Step 27: (Gwendolyn location unchanged; bobby pin location unchanged) {Gwendolyn location: closet; bobby pin location: closet's ledge; Gwendolyn sees bobby pin: True; Gwendolyn's belief on bobby pin: closet's ledge}
- Step 28: (Gwendolyn location unchanged; bobby pin location changed) {Gwendolyn location: closet; bobby pin location: pantry's ledge; Gwendolyn sees bobby pin: False; Gwendolyn's belief on bobby pin: closet's ledge}
- Step 29: (Gwendolyn location changed; bobby pin location unchanged) {Gwendolyn location: None; bobby pin location: pantry's ledge; Gwendolyn sees bobby pin: False; Gwendolyn's belief on bobby pin: closet's ledge}
... ...

## G.3 Countdown

Example G.4 shows an example of GPT-4o incorrectly updating search states for a 2K example of the countdown task. The model drops the branch of $36 * 6 = 216$ which is included in the ground truth solution.

---

**Example G.4: An example of incorrectly updating search states for Countdown**

**Problem**
You will be given four numbers and a target number, your task is to find a way to use all four numbers exactly once, along with the basic operations (+, -, *, /), to reach the target number.
Numbers: [30, 6, 8, 6]
Target: 27

---

**Ground Truth - Solution**
<Solution>
30 + 6 = 36
36 * 6 = 216
216 / 8 = 27
</Solution>

> **Model Output - Solution**
> No feasible solution found.
> **Model Output - Solving Procedure**
> ... ...
>   |- Try 30 + 6 = 36. Add 36 to the number set. Current number set: [36, 8, 6], target: 27. Options for choosing two numbers: [(36, 8), (36, 6), (8, 6)].
> ... ...
>     |- Pick two numbers (36, 6) (numbers left: [8]). Try possible operations.
>       |- Try 36 + 6 = 42. Add 42 to the number set. Current number set: [42, 8], target: 27, just two numbers left.
>         |- Try 42 + 8 = 50. Evaluate 50 != 27, drop this branch.
>         |- Try 42 - 8 = 34. Evaluate 34 != 27, drop this branch.
>         |- Try 42 * 8 = 336. Evaluate 336 != 27, drop this branch.
>         |- Try 42 / 8 = 5.2. 5.2 is a decimal, drop this branch.
>       |- Try 36 - 6 = 30. Add 30 to the number set. Current number set: [30, 8], target: 27, just two numbers left.
>         |- Try 30 + 8 = 38. Evaluate 38 != 27, drop this branch.
>         |- Try 30 - 8 = 22. Evaluate 22 != 27, drop this branch.
>         |- Try 30 * 8 = 240. Evaluate 240 != 27, drop this branch.
>         |- Try 30 / 8 = 3.7. 3.7 is a decimal, drop this branch.
>     |- Try 36 * 6 = 216. Evaluate 216 != 27, drop this branch.
> ... ...

## G.4 Travel Planning

Example G.5 shows an example of GPT-4o failing to update search states correctly of an 8K test set of the travel planning task. The model incorrectly drops the branch of [Mykonos, Zurich] after dropping one of the deeper child branches where the [Mykonos, Zurich] branch is part of the ground truth plan.

> **Example G.5: An example of search state update error for Travel Planning**
>
> **Problem**
> You plan to visit 9 European cities for 19 days in total. You only take direct flights to commute between cities. You want to spend 5 days in Stockholm. You have to attend a workshop in Stockholm between day 2 and day 6. You want to spend 4 days in Riga. You plan to stay in Manchester for 2 days. You plan to stay in Stuttgart for 2 days. From day 7 to day 8, there is a annual show you want to attend in Stuttgart. You would like to visit London for 3 days. You plan to stay in Reykjavik for 2 days. You plan to stay in Tallinn for 2 days. You want to spend 2 days in Santorini. You would like to visit Budapest for 5 days.
>
> Here are the cities that have direct flights:
> from Budapest to Manchester, from Stuttgart to London, from Reykjavik to London, from Reykjavik to Stuttgart, from Stuttgart to Manchester, from London to Santorini, from Stockholm to Manchester, from Stockholm to Riga, from Stockholm to Stuttgart, from Stuttgart to Budapest, from Riga to Tallinn, from London to Manchester, from Santorini to Manchester, from Stockholm to London, from Stockholm to Reykjavik, from Stockholm to Santorini, from Budapest to London, from Tallinn to Stockholm, from Reykjavik to Budapest, from Manchester to Riga.
>
> Find a trip plan of visiting the cities for 19 days by taking direct flights to commute between them.

> **Ground Truth - Plan**
> Here is the trip plan for visiting the 7 European cities for 18 days:
> **Day 1-5:** Arriving in Mykonos and visit Mykonos for 5 days.
> **Day 5:** Fly from Mykonos to Zurich.
> **Day 5-7:** Visit Zurich for 3 days.
> **Day 7:** Fly from Zurich to Manchester.
> **Day 7-8:** Visit Manchester for 2 days.
> **Day 8:** Fly from Manchester to Lisbon.
> **Day 8-10:** Visit Lisbon for 3 days.
> **Day 10:** Fly from Lisbon to Lyon.

**Day 10-13:** Visit Lyon for 4 days.
**Day 13:** Fly from Lyon to Barcelona.
**Day 13-14:** Visit Barcelona for 2 days.
**Day 14:** Fly from Barcelona to Prague.
**Day 14-18:** Visit Prague for 5 days.
**Model Output - Plan**
No feasible plan found.
**Model Output - Solving Procedure**
... ...
  |- Current day: 7. Current plan: [Mykonos, Zurich].
  |- Check whether the city with an arrival day of Day 7 - is fixed.
  |- No. Consider possible options from cities needing arrangement: [Lyon, Lisbon, Manchester]
and explore these options in order.
    |- Try arranging to visit Lyon from Day 7. Duration: 4 days. Schedule: Day 7 - 10.
      |- Check for direct flight from Zurich to Lyon.
      |- No. Drop this branch.
    |- Try arranging to visit Lisbon from Day 7. Duration: 3 days. Schedule: Day 7 - 9.
      |- Check for direct flight from Zurich to Lisbon.
      |- Yes.
      |- Check whether this arrangement is compatible with the next fixed schedule after Day 7:
Barcelona (Day 13 - 14).
      |- The departure day of Lisbon is Day 9. The arrival day of Barcelona is Day 13. Day 9 is not
later than Day 13. This arrangement is compatible.
      |- This arrangement is feasible for now. Continue to arrange the rest of the plan.
      |- Current day: 9. Current plan: [Mykonos, Zurich, Lisbon].
... ...
        |- Current day: 12. Current plan: [Mykonos, Zurich, Lisbon, Lyon].
        |- Check whether the city with an arrival day of Day 12 - is fixed.
        |- No. Consider possible options from cities needing arrangement: [Manchester] and explore
these options in order.
          |- Try arranging to visit Manchester from Day 12. Duration: 2 days. Schedule: Day 12 - 13.
            |- Check for direct flight from Lyon to Manchester.
            |- No. Drop this branch.
          |- Fail to arrange any option on day 12 in the current arrangement. Drop this branch.
        |- Fail to arrange any option on day 9 in the current arrangement. Drop this branch.
      |- Fail to arrange any option on day 7 in the current arrangement. Drop this branch.
    |- Fail to arrange any option on day 5 in the current arrangement. Drop this branch.
  |- Fail to arrange any option on day 1 in the current arrangement. Drop this branch.
  |- Try arranging to visit Manchester from Day 1. Duration: 2 days. Schedule: Day 1 - 2.
... ...

Example G.6 shows an example of GPT-4o hallucinating non-existent direct flights from an 8K example of travel planning. The model outputs a plan with a flight from Barcelona to Santorini at Day 5 while such a direct flight does not exist in the problem statement.

---

**Example G.6: An example of hallucinations for Travel Planning**

**Problem**
You plan to visit 7 European cities for 14 days in total. You only take direct flights to commute between cities. You want to spend 2 days in Santorini. During day 5 and day 6, you have to attend a conference in Santorini. You would like to visit Porto for 5 days. You plan to stay in Reykjavik for 3 days. You would like to visit Stockholm for 4 days. You want to spend 2 days in Barcelona. You want to spend 2 days in Dubrovnik. You plan to stay in Stuttgart for 2 days.

Here are the cities that have direct flights:
from Stockholm to Stuttgart, from Dubrovnik to Barcelona, from Barcelona to Porto, from Stuttgart to Porto, from Santorini to Barcelona, from Stockholm to Barcelona, from Reykjavik to Stuttgart, from Dubrovnik to Stockholm, from Stockholm to Reykjavik, from Stockholm to Santorini, from Barcelona to Stuttgart, from Barcelona to Reykjavik.

Find a trip plan of visiting the cities for 14 days by taking direct flights to commute between them.

---

**Ground Truth - Plan**
Here is the trip plan for visiting the 7 European cities for 14 days:
*Day 1-2:** Arriving in Dubrovnik and visit Dubrovnik for 2 days.
*Day 2:** Fly from Dubrovnik to Stockholm.
*Day 2-5:** Visit Stockholm for 4 days.
*Day 5:** Fly from Stockholm to Santorini.
*Day 5-6:** Visit Santorini for 2 days.
*Day 6:** Fly from Santorini to Barcelona.
*Day 6-7:** Visit Barcelona for 2 days.
*Day 7:** Fly from Barcelona to Reykjavik.
*Day 7-9:** Visit Reykjavik for 3 days.
*Day 9:** Fly from Reykjavik to Stuttgart.
*Day 9-10:** Visit Stuttgart for 2 days.
*Day 10:** Fly from Stuttgart to Porto.
*Day 10-14:** Visit Porto for 5 days.
**Model Output - Plan**
**Day 1-2:** Arriving in Dubrovnik and visit Dubrovnik for 2 days.
*Day 2:** Fly from Dubrovnik to Stockholm.
*Day 2-5:** Visit Stockholm for 4 days.
**Day 5:** Fly from Stockholm to Barcelona.
**Day 5:** Fly from Barcelona to Santorini.
*Day 5-6:** Visit Santorini for 2 days.
*Day 6:** Fly from Santorini to Barcelona.
*Day 6-7:** Visit Barcelona for 2 days.
*Day 7:** Fly from Barcelona to Reykjavik.
*Day 7-9:** Visit Reykjavik for 3 days.
*Day 9:** Fly from Reykjavik to Stuttgart.
*Day 9-10:** Visit Stuttgart for 2 days.
*Day 10:** Fly from Stuttgart to Porto.
*Day 10-14:** Visit Porto for 5 days.

## H   Details of Prompts

We list the evaluation prompt template for all the tasks in LONGPROC.

---

**Prompt H.1: Template for the HTML to TSV Task**

[TASK]
Your task is to extract specific information from an HTML webpage and output the extracted information in a tsv file. You will be first given an HTML webpage. Then, you should follow the specific instruction provided later and output the tsv file following the format provided in the instruction.

[INPUT WEBPAGE]
'''html
html page
'''

[TARGET INFORMATION]
Based on the HTML webpage above about {task_topic}, extract the following properties from the items listed on the webpage: target information

[OUTPUT FORMAT]
Structure your output in TSV format such that each row of your output corresponds to the aforementioned properties of an item and each property is separated from each other by a tab " ".
Your output should be in the following format:
'''tsv
tsv header
{{Your TSV output}}
'''

---

[IMPORTANT NOTES]
- Make sure that you have read through all items listed on the webpage and followed the same order as they appear on the webpage.
- If you are asked to only extract some rows that satisfy specific conditions, ONLY extract those rows that satisfy the conditions and do NOT include other irrelevant rows in your output.
- If a property of an item is blank, not applicable, or not parseable, please set the property to "N/A" for the item.
- If a property spans multiple lines, please extract all the lines and replace the newline character with a space character.
- If a property consists of a list of items, please replace the newline character with a space character and separate the items with a comma ",".
- If there are any special characters, numerical values of a specific format, or any unusual formatting in the property, please keep them as they are. If the property comes with a unit, please keep the unit as well in the property.
- Do not include html tags in the extracted information. Only include the text.
- Do not provide any additional information in your output other than the tsv.

Now, extract the information from the HTML webpage above and follow the output format above in your answer.

---

## Prompt H.2: Template for the Pseudocode to Code Task

[TASK]:
You will be given lines of pseudocode, your task is to write the corresponding C++ code. The pseudocode will provide detailed description of the c++ code line by line. The pseudocode is garanteed to be correct and complete.

[INSTRUCTION]:
The following libraries are already included in the code.
```cpp
#include <cstdio>
#include <iostream>
#include <vector>
#include <algorithm>
#include <numeric>
#include <cmath>
#include <cstring>
#include <set>
#include <map>
#include <queue>
#include <stack>
#include <list>
#include <fstream>
#include <climits>
#include <cassert>
#include <iomanip>
#include <sstream>
#include <bitset>
using namespace std;
```
Do not include them in your code again. Please surround your code with '''cpp and ''' markers. Note that the code should correspond to the pseudocode line by line.

[PSEUDOCODE]:
pseudocode

[CODE]:

---

**Prompt H.3: Template for the Path Traversal Task**

[TASK]
In a completely hypothetical world, there are a number of cities. Each city has a one-way connection to only one other city via a specific transit method (bus, train, plane, or ferry). Your task is to provide a route from a city to another city. You should follow the specific instruction provided later and output the route following the format provided in the instruction.

[IMPORTANT NOTES]
- All connections are one-way. If city A is connected to city B, you can travel from A to B, but not the other way around.
- Because each city is connected to only one other city, so there's only one possible route. To find the route, you can simply start from the starting city, identify the next city it's connected to, and repeat the process until you reach the destination city.
- Please follow the exact format specified below when outputting the route.

[OUTPUT FORMAT]
Please mark the route with <Route>and </Route>tags. The route should be in the following format, where one line is one step of the route:
<Route>
From <CITY_NAME>, take a <TRANSIT_METHOD>to <CITY_NAME>.
...
From <CITY_NAME>, take a <TRANSIT_METHOD>to <CITY_NAME>.
</Route>

[EXAMPLE]
In a hypothetical world, there are a number of cities. Each city has a one-way connection to only one other city via a specific transit method. The details of the cities are as follows:

Fort Worth is a lively city. You can travel from Fort Worth to Manchester by ferry.
Leeds is a lively city. You can travel from Leeds to London by bus.
Manchester is a lively city. You can travel from Manchester to Indianapolis by plane.
Houston is a lively city. You can travel from Houston to London by ferry.
Charlotte is a lively city. You can travel from Charlotte to Charlotte by bus.
London is a lively city. You can travel from London to San Antonio by train.
San Antonio is a lively city. You can travel from San Antonio to Kitchener by train.
Seattle is a lively city. You can travel from Seattle to London by train.
Indianapolis is a lively city. You can travel from Indianapolis to Houston by ferry.

Now find the route from Manchester to Kitchener based on the information above.

<Route>
From Manchester, take a plane to Indianapolis.
From Indianapolis, take a ferry to Houston.
From Houston, take a ferry to London.
From London, take a train to San Antonio.
From San Antonio, take a train to Kitchener.
</Route>

[PROBLEM]
`problem context`
Now find the route from `src city` to `dst city` based on the information above. Some reminders:
- All connections are one-way. You can solve the problem by iteratively finding the next city to travel to until you reach the destination city.
- Follow the specific format for the route output. Mark the route with <Route>and </Route>tags.

---

**Prompt H.4: Template for the Countdown Task**

[TASK]
You will be given four numbers and a target number, your task is to find a way to use all four numbers exactly once, along with the basic operations (+, -, *, /), to reach the target number.

---

[RULES]
- You can use each number exactly once.
- You can use the four basic operations (+, -, *, /).
- The intermediate results must be integers (no decimals allowed).
- The intermediate results must be positive.
- The intermediate results will not exceed 2000.

[APPROACH]
We will solve the problem by searching. Starting from a given set of four numbers, we will follow this search process:
- At each state, we will consider all possible number pairs (in order) from the current number set. Choose one pair and apply one of the four basic operations to them to obtain a new number.
* If there are still numbers left, we will add the new number to the number set and continue the search.
* If we have used all numbers, we will check if the new number is equal to the target number. If it is, we have found the solution. Otherwise, we will backtrack.
- Suppose the two numbers we choose are a and b (where a >= b). We will try the four options (a + b), (a - b), (a * b), (a / b) to obtain the new number. Remember to always use the larger number as the first operand.
- If the new number is a decimal, or exceeds the maximum intermediate result, we will discard this branch and backtrack.
- We will continue this process until we reach the target number with four numbers used or exhaust all possible combinations.

[EXAMPLES]
few shot examples

[Problem]
Now, solve the following problem. Note that:
- Please carefully read the approach and examples provided above, and follow them to solve the problem.
- Please ALWAYS include your search procedure. The search procedure should follow the format of the examples provided above.
- Please mark your answer with <Solution>and </Solution>tags. The solution should be a sequence of three equations exactly following the format of the examples above, with no additional text in between.

Numbers: numbers
Target: target

---

## Prompt H.5: Template for the Travel Planning Task

TASK:
Your task is to create a trip plan based on given constraints regarding cities to visit, duration of stays for each city, and available direct flight connections.

REQUIREMENTS AND NOTES:
- You will arrange a trip plan for visiting several cities for a specified total number of days.
- You will be informed about how long we will stay in each city. Some cities have fixed schedules because of pre-planned events. You have to follow the fixed schedules for those cities. Cities without fixed schedules need to be arranged according to the constraints.
- You will be provided with information about direct flight connections between cities. Only direct flights may be used to travel between cities. Note that the flight information is one-way. For example, if there is a direct flight from City A to City B, it does not necessarily mean that there is a direct flight from City B to City A, unless it is explicitly mentioned. There ALWAYS exists a direct flight from the starting point to the first city in the plan.
- When calculating the duration of a stay in a city, count both arrival and departure days as full days. If you arrive at a city on Day x and stay for y days, you will leave the city on Day x + y - 1. For example, if you arrive in a city on Day 1 and stay for 3 days, you will depart on Day 3.
- When handling the cities with fixed schedules, these fixed schedules may overlap with the arrival and departure days of other cities. For example, if City A has a fixed schedule from Day 4 to Day 7, you can depart another city and arrive in City A on Day 4, and you can depart City A and arrive in

another city on Day 7.

APPROACH:
We will solve the problem by searching. You will follow this process:
- First, read the constraints carefully and identify the cities that have fixed schedules and the cities needing arrangement.
- Next, you will search for the trip plan starting from Day 1. At each day, you will execute the following steps:
* Check whether the schedule for the current day is fixed. If it is fixed, the only option for the day is to follow the fixed schedule. If it is not fixed, you will consider possible options from the cities needing arrangement.
* For each option, you will check whether it is feasible to arrange the city on the current day. You will first check whether there is a direct flight from the last city on the plan to the current city.
* For each option, you will also check whether the arrangement is compatible with the fixed schedules. Recall that fixed schedules may overlap with the arrival and departure days of other cities. It is considered incompatible only if the departure day is later than the required arrival day for the following fixed schedule. For example, you can depart from City A on Day 3 and arrive in City B, which has a fixed schedule of Day 3 - 6. However, you cannot depart from City A on Day 4, which is later than the required arrival day for City B.
* If the arrangement is not feasible, you will drop this branch and try the next option. If the arrangement is feasible, you will continue to arrange the rest of the plan. If you fail to arrange any option on the current day, you will drop this branch and backtrack to the previous stage.
- You will continue this process until you have arranged all the cities in the trip plan and find a complete plan.
- Finally, you will output the complete trip plan.

EXAMPLES:
few shot examples

YOUR TASK:
Now, make a trip plan according to the following constraints. Note that:
- Please carefully read the approach and examples provided above, and follow them to make the trip plan.
- Please ALWAYS include your solving procedure and mark it with <Solving Procedure>and </Solving Procedure>tags. The solving procedure should follow the format of the examples provided above.
- Please mark your final plan with <Plan>and </Plan>tags. The final plan should also follow the format of the examples provided above.

problem description

