# OpenReview forum: "LongProc: Benchmarking Long-Context Language Models on Long Procedural Generation"
_colmweb.org/COLM/2025/Conference — COLM 2025_

### Official Review · Reviewer_kGCV · 2025-05-08

**Rating:** 7
**Confidence:** 5
**Ethics Flag:** 1

**Summary:**

The paper describes an analysis of LLMs for effective use of large contexts over different levels of sizes and access of the information in the contexts. The authors curated several datasets across tasks which involves long context like HTML to CSV. The tasks are created with different context lengths and are easily to be verified using simple matches rather than having to rely on an other LLM as evaluator. They performed a deep analysis over several LLMs and studied their performance across the tasks and the different sizes of context.

**Questions To Authors:**

Is the Pseudocode task really a sequential task? How do you verify that the generated code is in the same order as the pseudocode if you only check unit tests? The model reads the pseudocode from left to right (it is autoregressive), but after that it might not need to attend sequentially while generating tokens.
Maybe an investigation of the attention scores for some of the small open models might give more insights.
I might have missed this, but do you create the same task in 3 different levels (each having other context lengths) or are they different tasks?

**Reasons To Accept:**

Very interesting studies with useful findings. Detailed analysis over several models. Good spectrum of tasks and extensive analysis of the results.

**Reasons To Reject:**

While there a limit of what the authors could test, I would like to see how models perform with regard to the level of irrelevant or contradictory information in the context.

---

> ### Author Response · Authors · 2025-06-01
> **Thank your for your helpful feedback!**
>
> Thank you for your thoughtful comments and feedback! Please find our clarifications and answers to the questions below.
>
> ### **Clarification on Contributions**
>
> > In summary: The paper describes an analysis of LLMs for effective use of large contexts over different levels of sizes and access of the information in the contexts. The authors curated several datasets across tasks which involve long context like HTML to CSV. The tasks are created with different context lengths.
>
> Thank you for the summary. We would like to clarify that, in addition to evaluating how models access dispersed information in long contexts, a central focus of our benchmark is **long-form generation**. To support this, we configure tasks with varying maximum generation lengths, as explained in our response to Q1 below. We highlight long-form generation because it is a critical yet relatively under-explored capability in existing benchmarks.
>
> ### **Answers to the Questions**
>
> > **Q1:** How do you create the tasks with 3 different difficulty levels? Do they contain the same set of tasks? Do they contain the same base data points (each having other context lengths)?
>
> **A:** We create three difficulty levels corresponding to different output lengths across the same set of six tasks in our benchmark. To construct these levels, we select subsets of data points that require varying output lengths. As a result, each difficulty level includes different subsets of examples, but they all span the same set of tasks. Please refer to **Appendix B** for details on the construction process.
>
> > **Q2:** While there is a limit to what the authors could test, I would like to see how models perform with regard to the level of irrelevant or contradictory information in the context
>
> **A:** This is an interesting point, though we consider such analysis somewhat out of scope for our current focus on long-form outputs and synthesis of dispersed information.
>
> That said, one of our tasks (HTML to TSV) offers a partial investigation of this issue. It includes two settings: 1) extracting all, where the model must identify all relevant items (e.g., “all movies in the page”), and 2) filtering, where the model must extract a subset based on specific conditions (e.g., “all action movies in the page”). Please refer to Appendix B.2 for details on how this task is constructed.
> In the table below, we report several models’ performance on the HTML to TSV (8K) task, with and without filtering (in our work, we only report the aggregated number). We observe a substantial drop in the filtering setting, suggesting difficulty in handling irrelevant information. We will include this discussion in any future version, but we feel a full analysis of robustness to irrelevant context is beyond the scope of this work.
>
> | HTML_to_TSV (8K)   | Extract All (No Filter) | Filtering |
> |--------------------|-------------|--------|
> | Llama-3.1-70B      |        55.7 |   38.4 |
> | GPT-4o-2024-08     |        78.1 |   52.8 |
> | Gemini-1.5-pro-001 |        77.5 |   62.5 |
>
> > **Q3:** Is the Pseudocode task a sequential task? How do you verify that the generated code is in the same order as the pseudocode if you only check unit tests?
>
> **A:** Thank you for raising this. Our evaluation does not enforce strict adherence to the original ordering, as we focus on the functional correctness of the complete program (following the original setup in Kulal et al., 2019). However, we explicitly instruct models: “Note that the translated code should correspond to the pseudocode line by line.” In practice, we observe that models largely follow the intended order. Minor deviations (such as inserting syntactic elements like braces) can occur, but the translated programs generally follow the original ordering.

---

> > ### Comment · Reviewer_kGCV · 2025-06-09
> >
> > I want to thank the authors for their comments. This clarified some of the concerns/questions I had. I will increase my scores accordingly.

---

### Official Review · Reviewer_eTor · 2025-05-11

**Rating:** 8
**Confidence:** 4
**Ethics Flag:** 1

**Summary:**

This work builds a new benchmark LongProc for long-context processing of LLMs. This benchmarks allows to evaluate LLMs long-context processing capabitilies along following directions: 1) how well does it understand long structured input 2) how good it is at generating long structured output. It includes 6 tasks, that allow reliable evaluation. This is the first benchmark for long context that proposes to asses simultaneously how well LLM is with long inputs, long outputs, and reliable and objective evaluation of these outputs. Benhcmark proposes 3 different difficulty leves limiting the generations to 500, 2k or 8k tokens.  Authors analyse behavour of  23 existing LLMs on such benchmark, including strong closed source and various open-source (instruction-tuned and reasoning) models.   They provide some interesting insights in models behavour: 1) even the strongest  models still struggle at generating 8K coherent output 2) coherence of the output decreases with amount of generated tokens 3) larger models are usually better at handling long context 4) reasinong models are generally better than instruction-tuned models in LC handling.

**Reasons To Accept:**

- This sounds like a very interesting and well constructed benchmark to evaluate Long Context handling of LLMs.
- Design of the tasks of the benchmark allow for reliable and interpretable evaluation which often tricky when we want to evaluate long generations.
- The analysis of existing models is insightful.
- The paper is well written and easy to follow

**Reasons To Reject:**

- Like with any benchmark: it could be hacked, and the evaluate data could quickly leak in the new revisions of the models.
(But that does not cancel the quality of this work, and I do not think it is  really a reason to reject)

---

> ### Author Response · Authors · 2025-06-01
> **Thank your for your thoughtful comments!**
>
> We appreciate your thoughtful comments and feedback. Please find our answer below.
>
> > **Q1:** Like with any benchmark: it could be hacked, and the evaluated data could quickly leak in the new revisions of the models.
>
> **A:** This is a good point, and thank you for bringing it up! We acknowledge the broader challenge of preventing data contamination in benchmarks. At the same time, we believe our long-form generation evaluation offers some natural resilience to this issue: 1) our benchmark evaluates long outputs, which are harder to memorize compared to short-form answers. 2) several tasks involve solving procedures that can be programmatically generated, allowing us to produce ground-truth outputs as needed. In our final release, we plan to withhold ground-truth procedures for tasks where this is appropriate. Thank you again for the helpful suggestion.

---

> > ### Comment · Reviewer_eTor · 2025-06-09
> > **Thank you for your response**
> >
> > Dear authors.
> >
> > Thank you for your comment. This indeed makes sense.
> >
> > I keep my score and believe this work worth being shared with the COLM community.

---

### Official Review · Reviewer_5YMC · 2025-05-12

**Rating:** 8
**Confidence:** 4
**Ethics Flag:** 1

**Summary:**

This paper proposes a benchmark for evaluating long-context language models (LCLMs), LongProc, focusing on long procedural generation, testing LCLMs on their ability to integrate highly dispersed information and to generate long procedural text. LongProc not only requires long text understanding, as has been the main focus of existing benchmarks, but also long text generation of up to 8K tokens. However, evaluations can still rely on rule-based approaches as the generated texts are procedural, as opposed to the LongWriter benchmark where the generated open-ended texts are inherently subjective. Analysis shows how even strong closed-source models like GPT-4o and Gemini-1.5-pro struggle at generating output for 8K tasks. And while reasoning models outperform instruction-tuned models across length-based difficulty levels (0.5K, 2K, 8K), they usually fail to generate final solutions for the 8K tasks within the enforced 16K output token budget.

**Questions To Authors:**

1. In the "Benefits of procedural generation" subsection, there is a comparison between traditional ICL and ICL with procedure. What is the default evaluation mode used to produce the results in Table 3?

**Reasons To Accept:**

1. The proposed benchmark is highly relevant for long-context language models (LCLMs), and offer evaluations on diverse tasks including both long context understanding (converting HTML to TSV, pseudo-code to code), as well as long procedural generation for the reasoning tasks (theory-of-mind tracking, countdown and travel planning). The benchmark is also set with three difficulty levels based on the length of output tokens (0.5K, 2K and 8K), making it suitable for evaluating a range of models in terms of long understanding vs long generation.
2. The proposed benchmark relies on rule-based evaluation, given the procedural output, but can still be used to examine models' capabilities in multi-step reasoning and procedure following.
3. Thorough analysis on various models' performance.

**Reasons To Reject:**

I have no major reasons to reject the paper.

---

> ### Author Response · Authors · 2025-06-01
> **Thank your for your helpful feedback!**
>
> Thank you for your thoughtful comments and feedback! Please find our answers below.
>
> > **Q1:** What is the default evaluation mode used to produce the results in Table 3?
>
> **A:** The default evaluation mode is ICL with procedure. Our primary goal is to assess models’ capabilities in long-form generation given a specific procedure. We include the full procedure in the prompt, to emphasize the challenge of extended generation rather than procedural discovery. We will clarify this more explicitly in any future version. Thanks for the suggestion!

---

> > ### Comment · Reviewer_5YMC · 2025-06-10
> > **Thank you for the clarification**
> >
> > Dear Authors,
> >
> > Thank you for your response. I keep my score as is since I believe you did a good work.

---

### Official Review · Reviewer_yAzi · 2025-05-15

**Rating:** 7
**Confidence:** 4
**Ethics Flag:** 1

**Summary:**

The manuscript introduces **LONGPROC**, a six‑task benchmark specifically aimed at evaluating long‑context language models (LCLMs) in *long‑form procedural generation* scenarios.  Each task forces a model to (i) retrieve information scattered across ≥ 32 K‑token inputs, (ii) execute a deterministic multi‑step procedure, and (iii) emit structured outputs as long as **8 K tokens**.  Three difficulty tiers (0.5 K / 2 K / 8 K) are provided.
The authors benchmark **23** contemporary LCLMs (open and closed weight, instruction‑tuned and reasoning‑trained) and report that:

* open‑weight models fail already at 2 K outputs, whereas even GPT‑4o and Gemini‑1.5‑Pro collapse at 8 K;
* model scale is a stronger predictor of success than architecture choice;
* models distilled on long chain‑of‑thought traces (“reasoning” variants) consistently beat instruction‑only counterparts;
* generation quality degrades sharply in later output chunks, highlighting long‑range coherence limits.

Code, prompts and evaluation scripts will be released, supporting reproducibility.

**Questions To Authors:**

1. What are the licensing terms for any third‑party data (HTML pages, IMDB content) and will raw data be fully redistributed?
2. Why cap each task at ~100 items, and are there plans for a larger hidden test set to deter prompt contamination?
3. How sensitive are results to the asymmetric prompt/output budgets between reasoning and instruction models?
4. Have you tried long‑context fine‑tuned baselines (LongLoRA, Flash‑Attention‑extended) or retrieval/memory‑augmented approaches?
5. How robust is the rule‑based evaluation to innocuous formatting deviations?
6. What was the compute cost (GPU hours / USD) of running the full benchmark?

**Reasons To Accept:**

* **Evaluation gap filled** – shifts focus from short‑answer recall to *long procedural generation*, a demand in agents, code synthesis, and planning systems.
* **Diverse yet automatable task suite** – six tasks span data extraction, code generation, symbolic search and planning; deterministic rules enable objective, low‑variance scoring.
* **Comprehensive empirical study** – 23 models × 3 tiers supply a valuable snapshot of current LCLM limits, with segment‑wise error analyses offering actionable insights.
* **Clear, actionable findings** – shows today’s “128 K‑context” claims do not translate to 8 K coherent outputs; underscores benefits of reasoning‑centric training.
* **Commitment to open release** – dataset, prompts and grader will be open‑sourced, encouraging community adoption and extension.
* **Practical relevance** – tasks such as HTML‑to‑TSV extraction and travel planning mirror real‑world use cases.

**Reasons To Reject:**

* **Dataset scale & coverage** – only ≈ 100 instances per task; Pseudocode‑to‑Code lacks an 8 K tier and Travel Planning omits the 0.5 K tier, limiting statistical power and cross‑task comparability.
* **Synthetic / deterministic bias** – tasks provide the exact procedure in the prompt, conflating instruction following with intrinsic reasoning; real‑world noise and open‑ended generation are not tested.
* **Unequal prompt budgets** – reasoning models are allowed longer prompts/outputs than instruction models, which affecting comparisons' interpretation.
* **Missing baselines** – no retrieval‑augmented, memory‑augmented or long‑LoRA‑tuned baselines; results may change with such methods.
* **Limited statistical analysis** – no significance testing; human evaluation covers only 35 examples per task.

---

> ### Author Response · Authors · 2025-06-01
> **Thank your for your thoughtful comments!**
>
> We appreciate your thoughtful comments and detailed feedback. Please find our answers to the questions below.
> > **Q1:** The benchmark contains a limited number of examples per setting. Are there plans for a larger hidden test set to deter prompt contamination?
>
> **A:** Thank you for the suggestion. Expanding the test set is feasible for many of our benchmarks where data can be programmatically generated, with the exception of HTML to TSV and Pseudocode to Code, which require substantial human annotation. We will consider including a larger test set in the final release.
>
> We chose 100 examples per task for two main reasons. First, long-form generation evaluation incurs high computational costs (see answer to Q7). Second, since our benchmark spans 6 diverse tasks, results averaged over 100 examples per task are sufficient to reveal meaningful trends. For example, as shown in Table 3 of the paper, proprietary models exhibit statistically significant performance differences on the 8K token tasks.
>
> > **Q2:** The tasks provide exact procedures in the prompt, conflating instruction-following with reasoning. Real-world noise and open-ended generation are not tested.
>
> **A:** Thank you for raising this. We provide explicit procedures to highlight the challenges of extended generation. While we acknowledge that this setup overlaps with instruction-following, we evaluate instruction following specifically in the context of reasoning tasks. These tasks often require deducing intermediate steps while maintaining logical coherence across long sequences, and we believe these capabilities are essential prerequisites for complex reasoning.
>
> We do include tasks that contain real-world text (HTML to TSV and Pseudocode to Code). However, our focus is on tasks with reliable and automatic evaluation metrics. While we acknowledge the importance of open-ended generation, such tasks often lack deterministic evaluation, and we consider them out of scope for this work.
>
> > **Q3:** (Unequal prompt budget)  Why do reasoning and instruction-tuned models have asymmetric output budgets?
>
> **A:** Thank you for this question. We believe this is a suitable way for evaluating these two sets of models with distinct characteristics. We allow more tokens for reasoning models to accommodate additional thinking tokens, since otherwise these models do not finish generation and we cannot extract the solutions. By contrast, instruction tuned models do not effectively use additional tokens even if we allow them more tokens, as they terminate most of the time (typically >90% across tasks) before hitting the max number of tokens. We will add this clarification in any future version.
>
> > **Q4:** (Missing baselines) How does long‑context fine‑tuned baselines (LongLoRA) or retrieval/memory‑augmented approaches perform?
>
> **A:** We would like to clarify that our evaluation includes long-context fine-tuned models. We believe that the models we evaluate represent the current state-of-the-art in long-context processing. Many of them are explicitly trained with long-context capabilities (e.g., as described in the Llama3 and Qwen2.5 technical reports). We also include ProLong, an open-source long-context model. These models achieve strong performance on long-context recall benchmarks  (>50.0 at 32K on RULER). By contrast, LongLoRA achieves only 29.4 at 32K on RULER, indicating it is less capable than the models we include.
>
> We do not evaluate retrieval-augmented or memory-augmented approaches because our tasks require using the entire context. These approaches typically involve chunking the input and selecting a subset of the context during generation, which would often result in task failure under our setup. Therefore, we consider these methods to be beyond the scope of our current evaluation.
>
> > **Q5:** How robust is the rule-based evaluation to innocuous formatting deviations?
>
> **A:** For each task, we implement specific normalization procedures to accommodate slight variations in formatting and wording. For instance, we use standard normalization steps used in QA evaluation (lowercasing, removing punctuation, etc,) for HTML to TSV tasks. We report the full details in **Appendix C**.
>
> > **Q6:** What are the licensing terms for any third-party data (e.g., HTML pages, IMDB content), and will raw data be fully redistributed?
>
> **A:** All third-party data is sourced from publicly available, open-source datasets released under redistribution-compliant licenses (e.g., CC 4.0).
>
> > **Q7:** What was the compute cost (GPU hours / USD) of running the full benchmark?
>
> **A:** For ~10B-scale models, we used a single H100 80GB GPU for approximately 9 GPU hours per model. For ~70B-scale models, we used 4×H100 GPUs with a total compute time of around 110 GPU hours. We will include these details in the revision. Thanks for your suggestions!

---

### Decision · Program_Chairs · 2025-07-08

**Decision:**

Accept

**Comment:**

This is an interesting paper that introduce a benchmark for evaluating long-context language models on procedural generation tasks. Overall, the reviewers are positive on this paper, and I share the same sentiment. The reviewers commend its practical relevance, diverse tasks, and insightful analysis across 23 models. While concerns were raised about dataset scale, missing baselines, and procedural vs. reasoning distinctions, the authors addressed most points thoughtfully. The authors should incorporate their responses and address reviewers' comment in the final cam-ready version of this paper. I recommend acceptance.